# Conserved mechanisms of microtubule-stimulated ADP release, ATP binding, and force generation in transport kinesins

**Joseph Atherton[1], Irene Farabella[1], I-Mei Yu[2], Steven S Rosenfeld[3], Anne Houdusse[2], Maya Topf[1], Carolyn A Moores[1]***

[1]Institute of Structural and Molecular Biology, Department of Biological Sciences, Birkbeck College, University of London, London, United Kingdom; [2]Structural Motility, Institut Curie, Centre National de la Recherche Scientifique, Paris, France; [3]Department of Cancer Biology, Lerner Research Institute, Cleveland Clinic, Cleveland, United States

**Abstract** Kinesins are a superfamily of microtubule-based ATP-powered motors, important for multiple, essential cellular functions. How microtubule binding stimulates their ATPase and controls force generation is not understood. To address this fundamental question, we visualized microtubule-bound kinesin-1 and kinesin-3 motor domains at multiple steps in their ATPase cycles—including their nucleotide-free states—at ~7 Å resolution using cryo-electron microscopy. In both motors, microtubule binding promotes ordered conformations of conserved loops that stimulate ADP release, enhance microtubule affinity and prime the catalytic site for ATP binding. ATP binding causes only small shifts of these nucleotide-coordinating loops but induces large conformational changes elsewhere that allow force generation and neck linker docking towards the microtubule plus end. Family-specific differences across the kinesin–microtubule interface account for the distinctive properties of each motor. Our data thus provide evidence for a conserved ATP-driven mechanism for kinesins and reveal the critical mechanistic contribution of the microtubule interface.

**\*For correspondence:** c.moores@mail.cryst.bbk.ac.uk

**Competing interests:** The authors declare that no competing interests exist.

**Reviewing editor**: Anthony A Hyman, Max Planck Institute of Molecular Cell Biology and Genetics, Germany

## Introduction

Kinesins are a large family of microtubule (MT)-based motors that play important roles in many cellular activities including mitosis, motility, and intracellular transport (*Vale, 2003*; *Hirokawa and Noda, 2008*; *Hirokawa et al., 2010*). Their involvement in a range of pathological processes also highlights their significance as therapeutic targets and the importance of understanding the molecular basis of their function (*Mandelkow and Mandelkow, 2002*; *Greber and Way, 2006*; *Henry et al., 2006*; *Stokin and Goldstein, 2006*; *Liu et al., 2012b*). Kinesins are defined by their motor domains that contain both the MT and ATP binding sites. Three ATP binding motifs—the P-loop, switch I, switch II–are highly conserved among kinesins (*Sablin et al., 1996*), myosin motors, and small GTPases (*Vale, 1996*). Kinesins also share a conserved mode of MT binding (*Woehlke et al., 1997*; *Alonso et al., 1998*) such that MT binding, ATP binding, and hydrolysis are functionally coupled for efficient MT-based work.

A number of kinesins drive long distance transport of cellular cargo (*Hirokawa et al., 2010*; *Soppina et al., 2014*) with dimerisation allowing them to take multiple 8 nm ATP-driven steps toward MT plus ends (*Svoboda et al., 1993*). Their processivity depends on communication between the two motor domains, which is achieved via the neck linker that connects each motor domain to the dimer-forming coiled-coil (*Hackney, 1994*; *Rice et al., 1999*; *Tomishige and Vale, 2000*; *Clancy et al., 2011*). In the

**eLife digest** The interior of a cell is a hive of activity, filled with proteins and other items moving from one location to another. A network of filaments called microtubules forms tracks along which so-called motor proteins carry these items. Kinesins are one group of motor proteins, and a typical kinesin protein has one end (called the 'motor domain') that can attach itself to the microtubules. The other end links to the cargo being carried, and a 'neck' connects the two. When two of these proteins work together, flexible regions of the neck allow the two motor domains to move past one another, which enable the kinesin to essentially walk along a microtubule in a stepwise manner.

To take these steps along microtubules, each kinesin motor domain in the pair must undergo alternating cycles of tight association and release from their tracks. This cycle is coordinated by binding and breaking down a molecule called ATP, which also provides the energy needed to take the next step. How the cycle of loose and tight microtubule attachment is coordinated with the release of the breakdown products of ATP, and how the energy from the ATP molecule is converted into the force that moves the motor along the microtubule, has been unclear.

Atherton et al. use a technique called cryo-electron microscopy to study—in more detail than previously seen—the structure of the motor domains of two types of kinesin called kinesin-1 and kinesin-3. Images were taken at different stages of the cycle used by the motor domains to extract the energy from ATP molecules. Although the two kinesins have been thought to move along the microtubule tracks in different ways, Atherton et al. find that the core mechanism used by their motor domains is the same.

When a motor domain binds to the microtubule, its shape changes, first stimulating release of the breakdown products of ATP from the previous cycle. This release makes room for a new ATP molecule to bind. The structural changes caused by ATP binding are relatively small but produce larger changes in the flexible neck region that enable individual motor domains within a kinesin pair to co-ordinate their movement and move in a consistent direction. This mechanism involves tight coupling between track binding and fuel usage and makes kinesins highly efficient motors.

The structures uncovered by Atherton et al. reveal a mechanism that links microtubule binding, the energy supplied to the motor domain and the force that moves the kinesin along a microtubule. Future work will clarify whether the key features observed in the motor domains of kinesin-1 and kinesin-3 are also found in other types of kinesin motors.

presence of MTs, ATP binding stimulates neck linker association (docking) with the motor domain towards the MT plus end, while ATP hydrolysis and MT release cause neck linker undocking (*Rice et al., 1999*; *Vale and Milligan, 2000*; *Skiniotis et al., 2003*; *Asenjo et al., 2006*); thus, the neck linker is required for both intra-dimer communication and directionality. However, even when the role of the motor N-terminus in reinforcing neck linker movement via cover neck bundle (CNB) formation is considered (*Hwang et al., 2008*; *Khalil et al., 2008*), the contribution of neck linker docking to the force generating mechanism(s) of these kinesins remains uncertain (*Rice et al., 1999*; *Vale and Milligan, 2000*; *Rice et al., 2003*). New insights into the conformational rearrangements of these motors when bound to MTs are essential to reveal how they produce force.

The high resolution X-ray structures of a range of kinesin motor domains have established a major communication route from the nucleotide binding site via helix-α4 (the so-called relay helix) to the neck linker, such that alternate conformations of helix-α4 either block or enable neck linker docking (*Vale and Milligan, 2000*; *Kikkawa et al., 2001*). However, the neck linker conformation seen in these MT-free structures is not always correlated to the nucleotide bound (*Vale and Milligan, 2000*; *Grant et al., 2007*). Cryo-electron microscopy (cryo-EM) has played a major role in elucidating several aspects of MT-bound kinesin mechanochemistry (*Sosa and Milligan, 1996*; *Sosa et al., 1997*; *Rice et al., 1999*; *Skiniotis et al., 2003*; *Hirose et al., 2006*; *Kikkawa and Hirokawa, 2006*; *Sindelar and Downing, 2007, 2010*; *Goulet et al., 2012, 2014*). Despite these contributions, and despite recent advances in the study of kinesin–tubulin complexes using X-ray crystallography (*Gigant et al., 2013*), several outstanding questions concerning kinesin mechanochemistry remain. Specifically, the mechanism by which MT binding stimulates the kinesin ATPase and in particular enhances Mg-ADP release by several orders of magnitude is not clear (*Hackney, 1988*; *Ma and Taylor, 1997*; *Sindelar, 2011*).

Although several speculative models have been proposed, an unambiguously interpretable structure of nucleotide-free MT-bound kinesin is currently lacking and is clearly critical in establishing how such transitions are achieved. Such a structure would also provide key insights into how ATP binding is coupled to both neck linker docking and force generation.

To address these major questions, we describe the MT-bound mechanochemical cycles of two plus-end directed human kinesin motor domains, a kinesin-1 (Kif5A) and a kinesin-3 (Kif1A), using cryo-EM structure determination at subnanometer resolution. Kinesin-1s (Kin1) and kinesin-3s (Kin3) are both important neuronal plus-end directed transport motors (*Hirokawa et al., 2009b*), but recent data suggest that Kin3 rather than Kin1 motors specifically are involved in long distance transport (*Soppina et al., 2014*). Their motor domains share 41% sequence identity, but profoundly different mechanochemistries—in which Kin1 dimers take processive steps and Kin3 monomers diffuse along MT tracks—have been proposed for these motors (*Hirokawa et al., 2009a*; *Sindelar, 2011*). Thus, we wanted to investigate these differences and compare the motors side by side. The high quality of our reconstructions, coupled to flexible fitting, enables new insights into the kinesin mechanism. In particular, nucleotide-free reconstructions for both motor domains reveal a conserved mechanism, whereby MT binding stimulates changes at the nucleotide-binding site favouring Mg-ADP release and conformationally primes the motor to receive Mg-ATP. We also show that relatively small structural transitions occur at the nucleotide-binding site on Mg-ATP binding, but that these lead to larger scale conformational changes and neck linker docking. Structural analysis of two different transport kinesins allows a direct comparison of their conserved mechanochemical features and identification of attributes that confer distinctive properties on each motor.

## Results

### MT-bound Kin1 and Kin3 reconstructions: an overview

We calculated MT-bound Kin3 reconstructions and pseudo-atomic models in four different nucleotide states: (1) Mg-ADP; (2) no nucleotide (NN), using apyrase treatment; (3) Mg-AMPPNP (a non-hydrolysable ATP analogue); and (4) Mg-ADPAlFx (an ATP hydrolysis transition state mimic), consistent with the previously described tight association of the Kin3 motor domain with MTs throughout its ATPase cycle (*Tables 1 and 2*, *Figure 1—figure supplements 1,2*; *Okada and Hirokawa, 2000*). We also calculated three Kin1 reconstructions and pseudo-atomic models: (1) no nucleotide (NN), (2) Mg-AMPPNP, and (3) Mg-ADPAlFx (*Tables 1 and 2*, *Figure 1—figure supplements 1,2*). Steady-state ATPase activities of the proteins that we used for our cryo-EM reconstructions (*Table 3*) show that the catalytic turnover of these motors are similar, but that the $K_m$MT of Kin3 is ~250× lower than Kin1. These values are broadly consistent with previous reports and also with our ability to form complexes for structure determination (*Woehlke et al., 1997*; *Okada and Hirokawa, 1999*; *Sindelar and Downing, 2010*). The conformations of both Kin3 and Kin1 in Mg-AMPPNP and Mg-ADPAlFx states were indistinguishable from

**Table 1.** Data set size and estimated reconstruction resolutions

| Kinesin and nucleotide state | Number of AU | FSCt 0.5 (0.143) | FSCtrue 0.5 (0.143) | Rmeasure 0.5 (0.143) | EMDB accession number |
|---|---|---|---|---|---|
| Kin3-Mg-ADP | 181,311 | 7.9 (6.3) | 8 (7) | 8.1 (7.5) | EMD-2768 |
| Kin3-NN | 187,538 | 7.4 (6.3) | 7.5 (6.3) | 7.8 (6.9) | EMD-2765 |
| Kin3-Mg-AMPPNP | 97,877 | 8.1 (6.9) | 8.2 (7.0) | 8 (7.3) | EMD-2766 |
| Kin3-Mg-ADPAlFx | 156,845 | 7.9 (6.8) | 8.3 (7.0) | 8 (7.3) | EMD-2767 |
| Kin1-NN | 168,974 | 8.2 (7.2) | 8.3 (7.4) | 8.3 (7.3) | EMD-2769 |
| Kin1-Mg-AMPPNP | 186,329 | 7.3 (6.0) | 7.5 (6.5) | 7.7 (6.9) | EMD-2770 |
| Kin1-Mg-ADPAlFx | 65,572 | 9 (7.3) | 9.1 (7.7) | 9.1 (8.1) | EMD-2771 |

For each reconstruction, the motor domain and nucleotide state, number of asymmetric units (AU) in the final reconstruction, the resolutions at a cut-off of 0.5 and 0.143 estimated by standard FSC (FSCt) and that corrected with the HRnoise substitution test (FSCtrue) (*Chen et al., 2013*) and by Rmeasure (*Sousa and Grigorieff, 2007*) and the EMDB accession number are given.

**Table 2.** Calculation of pseudo-atomic models

| Kinesin and nucleotide state | Models used | CCC initial model | CCC final model | PDB code |
|---|---|---|---|---|
| Kin3-Mg-ADP | 1VFZ (*Nitta et al., 2004*) | 0.66 | 0.68 | 4uxs |
| | 1I5S (*Kikkawa et al., 2001* | | | |
| | 4AQW (*Goulet et al., 2012*) | | | |
| Kin3-NN | 1VFZ/1I5S/4HNA (*Gigant et al., 2013*)/4AQW | 0.63 | 0.68 | 4uxo |
| Kin3-Mg-AMPPNP | 1VFV (*Nitta et al., 2004*) | 0.72 | 0.75 | 4uxp |
| | 4HNA | | | |
| Kin3-Mg-ADPAlFx | 1VFV/4HNA | 0.74 | 0.75 | 4uxr |
| Kin1-NN | 1BG2 (*Kull et al., 1996*)/4HNA/4AQW | 0.71 | 0.73 | 4uxt |
| Kin1-Mg-AMPPNP | 4HNA | 0.73 | 0.76 | 4uxy |
| Kin1-Mg-ADPAlFx | 4HNA | 0.69 | 0.72 | 4uy0 |

A set of starting models were used for each nucleotide state of each motor. Flexible fitting and further refinement were performed using Flex-EM and Modeller (see 'Materials and methods'). Global CCCs of models with their respective reconstructions were calculated using the *Fit In Map* tool in Chimera. PDB accession codes for the final models are also shown.

**Table 3.** Steady-state MT-activated ATPase parameters of our Kin3 and Kin1 motor domain constructs

| | Kin3 (Kif1A) | Kin1 (Kif5A) |
|---|---|---|
| $k_{cat}$ (s$^{-1}$) | 43.4 ± 1.0 | 34.2 ± 5.7 |
| $K_{0.5ATP}$ (µM) | 30 ± 10 | 25 ± 5 |
| $K_{0.5\ MT}$ (nM) | 53.7 ± 5.7 | 12,745 ± 4041 |

each other at the resolution of our reconstructions (global RMSD: Kin3 ADPAlFx/AMPPNP = 0.7 Å; Kin1 ADPAlFx/AMPPNP = 0.6 Å) as had been previously shown in other studies of transport kinesins (Kif5B; *Sindelar and Downing, 2010*; *Gigant et al., 2013*). Thus, for simplicity, we describe here one Mg-ATP-analogue ('Mg-ATP-like') reconstruction for each kinesin (Kin3: Mg-ADPAlFx; Kin1: Mg-AMPPNP). Views of the alternative Mg-ATP-like reconstructions for each kinesin are shown in figure supplements.

All our reconstructions have, as their asymmetric unit, a triangle-shaped motor domain bound to an αβ-tubulin dimer within the MT lattice (*Figure 1*). The structural comparisons below are made with respect to the MT surface, which, at the resolution of our structures (~7 Å, *Table 1*), is the same (CCC > 0.98 for all). As is well established across the superfamily, the major and largely invariant point of contact between kinesin motor domains and the MT is helix-α4, which lies at the tubulin intradimer interface (*Figure 1C*, *Kikkawa et al., 2001*). However, multiple conformational changes are seen throughout the rest of each domain in response to bound nucleotide (*Figure 1D*). Below, we describe the conformational changes in functionally important regions of each motor domain starting with the nucleotide-binding site, from which all other conformational changes emanate.

## MT binding drives Mg-ADP release and primes the nucleotide-binding site to respond to Mg-ATP binding

The nucleotide-binding site (*Figure 2*) has three major elements: (1) the P-loop (brown) is visible in all our reconstructions; (2) loop9 (yellow, contains switch I) undergoes major conformational changes through the ATPase cycle; and (3) loop11 (red, contains switch II) that connects strand-β7 to helix-α4, the conformation and flexibility of which is determined by MT binding and motor nucleotide state. The presence or absence of density for nucleotide in the nucleotide-binding site in each reconstruction (*Figure 2* and *Figure 2—figure supplement 5*) is consistent with the well-established sample preparation methods used (see 'Materials and methods'). In the Kin3-Mg-ADP reconstruction, the N-terminal half of helix-α4 lies at the back of the nucleotide-binding site where its N-terminal end is partially flexible (*Figure 2A*). ~50% of the adjacent loop11 is not visible presumably also due to flexibility, and density for this loop is only visible close to the P-loop at the edge of the motor's central β-sheet. In

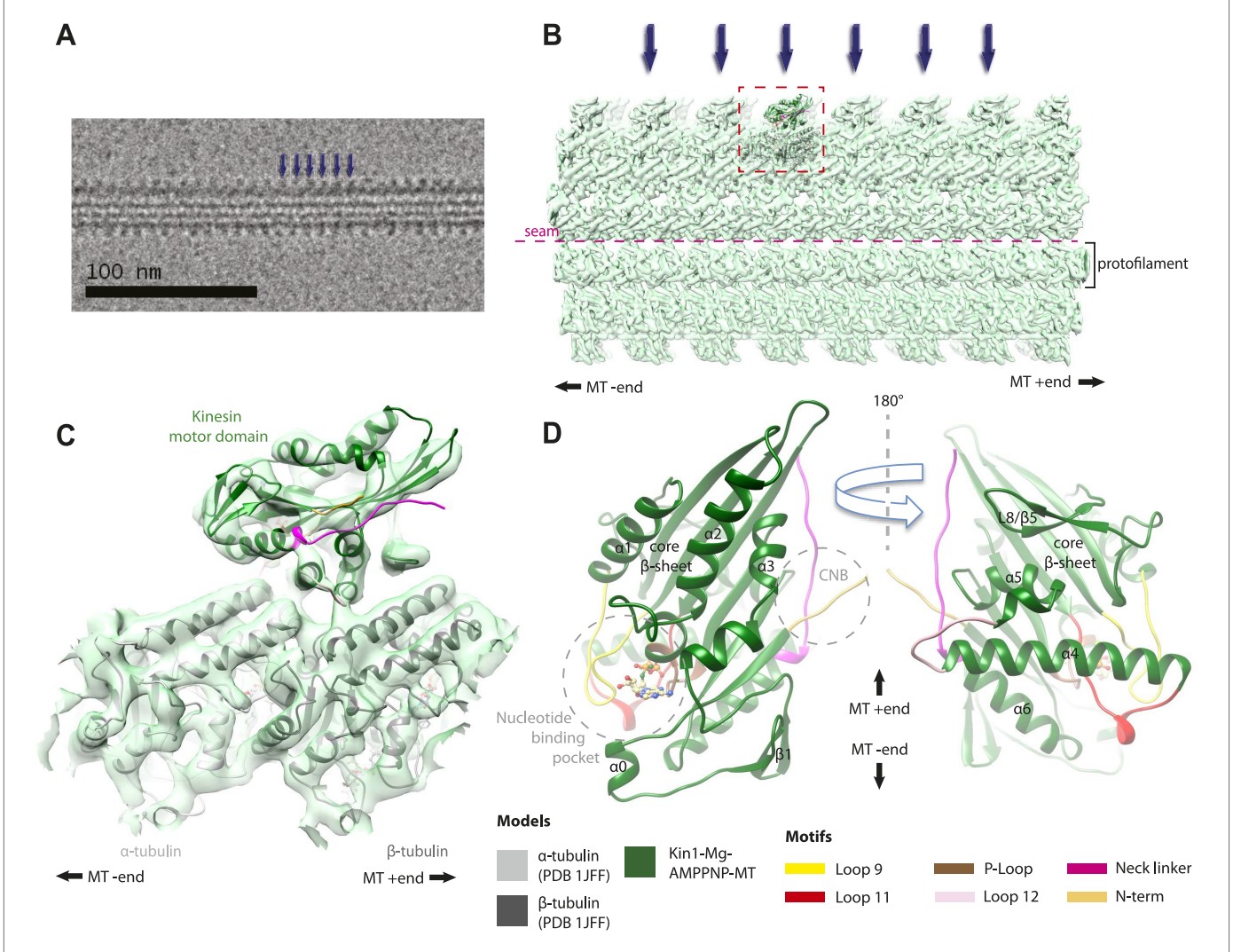

**Figure 1**. Overview of MT-bound kinesin motor domain cryo-EM reconstructions. (**A**) Example cryo-EM image of kinesin-decorated MT (Kin1-Mg-AMPPNP); blue arrows indicate individual Kin1 motor domains. (**B**) Example of cryo-EM reconstruction of 13 protofilament, kinesin-decorated MT (Kin1-Mg-AMPPNP); blue arrows indicate individual Kin1 motor domains, and the dotted red box shows an asymmetric unit. A single protofilament is indicated along with the position of the lattice seam. (**C**) Example of an individual asymmetric unit (Kin1-Mg-AMPPNP), contoured to show secondary structural elements. (**D**) Two views, related by 180°, of an exemplar pseudo-atomic model (Kin1-Mg-AMPPNP) calculated using our cryo-EM reconstruction. The major mechanochemical elements discussed in the text are colour-coded as indicated in the key.

The following figure supplements are available for figure 1:

**Figure supplement 1**. Resolution estimation for cryo-EM reconstructions.

**Figure supplement 2**. Local assessment of fit quality of the pseudo-atomic models within the cryo-EM density.

contrast, density corresponding to loop9 is clearly defined: the 4-turn helix-α3 is broken by a single residue, before two further helical segments are seen, one of which coordinates Mg-ADP, together with switch II (*Coureux et al., 2003*; *Hirose et al., 2006*; *Kull and Endow, 2013*). The conformations of loop9 and loop11 in this reconstruction are thus essentially the same as is seen in the Kin3-Mg-ADP crystal structure (*Kikkawa et al., 2001*).

In the Kin3-NN reconstruction (*Figure 2B*), the N-terminus of helix-α4 is fully stabilised, while the C-terminal portion of loop11 adopts a helical turn that forms a new contact with α-tubulin that likely contributes to the strengthened motor domain-MT interaction in the NN state (*Nakata and Hirokawa,*

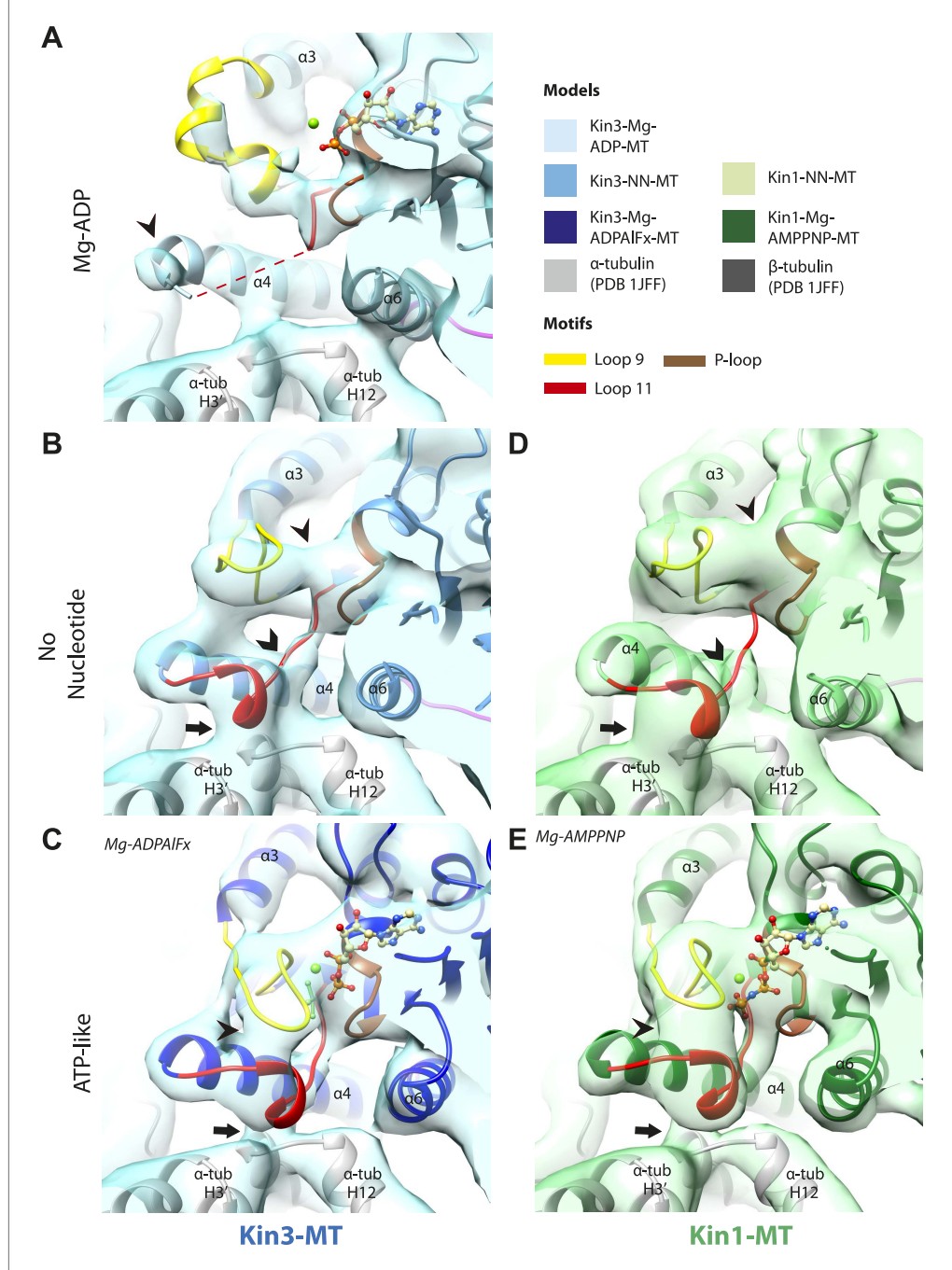

**Figure 2**. Conserved conformations at the nucleotide-binding pocket in Kin3s and Kin1s. (**A**–**C**) The nucleotide-binding pocket of MT-bound Kin3 reconstructions (shown as blue transparent density) in (**A**) Mg-ADP, model shown in light blue; the arrowhead indicates residual flexibility in the helix-α4 N-terminus and the region of loop11 for which density is missing is depicted by a dotted red line; (**B**) no nucleotide (NN), model shown in mid-blue; density connects the C-terminal helical turn of loop11 with the MT (arrow), density corresponding to the rest of loop11 is seen (chevron) and density now connects the extended loop 9 and the P-loop (arrowhead); (**C**) Mg-ADPAlFx, model shown in dark blue; the C-terminal helical turn of loop11 has moved away from the MT (arrow) and strong density is seen connecting it, helix-α4 and loop9 around the bound nucleotide. (**D**–**E**) The nucleotide-binding pocket of MT-bound Kin1 reconstructions (shown as green transparent density) in (**D**) no nucleotide, model shown in light green; density connects the C-terminal helical turn of loop11 with the MT (arrow), density corresponding to the majority of loop11 is seen (chevron) and density now connects the extended loop 9 and the P-loop (arrowhead); (**E**) Mg-AMPPNP, model shown in dark green; the C-terminal helical turn of loop11 has moved away from the MT
*Figure 2. Continued on next page*

*Figure 2. Continued*

(arrow) and strong density is seen connecting it, helix-α4 and loop9 around the bound nucleotide. In all reconstructions, density for the motor domain was contoured to an equivalent volume.

The following figure supplements are available for figure 2:

**Figure supplement 1**. Conserved conformations at the nucleotide-binding pocket in Kin3 and Kin1 alternative ATP-like states.

**Figure supplement 2**. Coordination of Mg-ADP cluster by loop9 and loop11.

**Figure supplement 3**. Conserved residues involved in MT-mediated stimulation of Mg-ADP release.

**Figure supplement 4**. Structural routes of communication between the nucleotide-binding pocket and helix-α6 for mechanochemical coupling.

**Figure supplement 5**. Occupancy of the nucleotide pocket.

*1995*). Density corresponding to the rest of loop11 is now also fully visible, such that switch II is seen running from the β-sheet core past the P-loop. Loop9 has undergone a large conformational change: helix-α3 now terminates after four turns and the resulting elongated conformation of loop9 forms a finger-like extension that reaches towards the nucleotide pocket and the new helical turn in loop11. Density connects this extended form of loop9 and the N-terminus of helix-α4; density also connects the P-loop and loop9 (as previously described for Kif5B; *Sindelar and Downing, 2007*; *Sindelar, 2011*). The Kin1-NN reconstruction shows a very similar configuration at the nucleotide-binding site (*Figure 2D*). This arrangement of the nucleotide–binding loops in both motors is striking because even in the absence of bound nucleotide, the loops adopt a conformation related (but not identical) to that formed when Mg-ATP is bound (*Parke et al., 2010*; *Chang et al., 2013*; *Gigant et al., 2013*). That is, MT-stimulated Mg-ADP release appears to conformationally prime the switch loops for Mg-ATP binding. The similarity of these reconstructions supports the idea of a conserved mechanism of: (1) MT-induced Mg-ADP release (*Figure 2—figure supplement 3*) and (2) MT priming of the conformation of the nucleotide-binding pocket to receive Mg-ATP in both Kin1s and Kin3s.

Because of this conformational priming, structural changes in the nucleotide-binding site upon ATP-binding are comparatively small when the NN and Mg-ATP-reconstructions are compared (*Figure 2B–E*, *Figure 2—figure supplement 1*). In both Kin3 and Kin1, loop9 now reaches further into the nucleotide-binding pocket to cradle the Mg-ATP mimic, enclosing it in a catalytically competent conformation and forming continuous density with the nucleotide and P-loop (*Figure 2C,E*). The C-terminus of loop11 retains a helical turn conformation similar to that observed in the nucleotide free reconstructions. Density for the N-terminus of loop11 runs from the core β-sheet past the P-loop and the γ-phosphate mimic. Importantly, however, in comparison to the nucleotide-free reconstruction, the loop11 helical turn shows reduced contact with tubulin and has moved toward loop9 and helix-α6 (see arrow, *Figure 2C,E*). The 'pincer-like' movement of the switch loops is associated with formation of a prominent connection of density between them and is consistent with a 'phosphate tube' structure similar to that described recently for other kinesins (*Parke et al., 2010*; *Sindelar and Downing, 2010*; *Chang et al., 2013*; *Gigant et al., 2013*). We note that, although the structure of the mammalian Kin1 Kif5A bound to MT has not previously been determined, our Kif5A reconstruction displays the major features seen in the recently published tubulin dimer-bound Kif5B Mg-ADPAlFx X-ray structure and to previous Mg-ATP analogue Kif5B cryo-EM reconstructions (*Sindelar and Downing, 2007*, *2010*; *Gigant et al., 2013*). Overall, in response to the presence of γ-phosphate, loop9 and loop11 draw closer to each other and to helix-α6 in both motors. This movement also reduces the density that connects loop11 with the MT.

## Movement and extension of helix-α6 controls neck linker docking

As shown in *Figure 2*, the N-terminus of helix-α6 is closely associated with elements of the nucleotide-binding site suggesting that its conformation alters in response to different nucleotide states. In addition, because the orientation of helix-α6 with respect to helix-α4 controls neck linker docking

(*Vale and Milligan, 2000*; *Kikkawa et al., 2001*), and because helix-α4 is held against the MT during the ATPase cycle, conformational changes in helix-α6 control movement of the neck linker.

In the Kin3-Mg-ADP reconstruction, helix-α6 contacts α-tubulin as was previously reported (*Figure 3A*, arrowhead; *Kikkawa and Hirokawa, 2006*); this interaction is likely to involve basic residues conserved in Kin3 (discussed below) and negatively charged residues in the N-terminal region of α-tubulin H12. The small β-sheet composed of strands-β1a,b,c (β-sheet1$_{abc}$) lies on top of helix-α6 and above the MT surface; this β-sheet is situated roughly perpendicular to the core β-sheet of the motor domain and contains the characteristically extended Kin3 loop2. In the Kin3-Mg-ADP state, the orientation of helix-α6 with respect to helix-α4 ensures both that helix-α6 cannot fully extend and the neck linker is undocked; this is indicated, first, by a lack of density between helix-α4 and helix-α6, and second by a lack of density along the core β-sheet (*Figure 3—figure supplement 3A*). The neck linker is mainly invisible and presumably disordered, consistent with previous reports (*Rice et al., 1999*; *Skiniotis et al., 2003*). However, some density that probably corresponds to the N-terminus of the neck linker is visible extending from the C-terminus of helix-α6, suggesting its flexible conformations are directed largely towards the MT minus end (*Figure 3A*, arrow and *Figure 3—figure supplement 3A*). Density that is likely to correspond to the Kin3 N-terminus is also visible but no single conformation can be distinguished.

In the Kin3-NN reconstruction, contact between helix-α6 and α-tubulin remains fixed, although the C-terminal end of helix-α4 is disconnected from the MT at its junction with the helix-α6 C-terminus (*Figure 3B*). The relative orientation of these helices ensures that the neck linker remains undocked and flexible; this is again indicated by the gap separating these helices and by density extending from the C-terminus of helix-α6, similar to that described in the Mg-ADP state (*Figure 3B* and *Figure 3—figure supplement 3B*). The flexible distribution of the N-terminus is also unaltered. The Kin1-NN reconstruction shows an overall similar configuration in the region of helix-α6, with its neck linker undocked and flexible and its N-terminus disordered (*Figure 3D* and *Figure 3—figure supplement 3E*). However, some family specific differences are apparent, both within the motor domain structure and at the motor–MT interface (*Figure 3D*). For example, in Kin1 β-sheet1$_{abc}$ appears more compact than in Kin3 because loop2 and loop3 are shorter. In Kin1 helix-α6, differences are present in the charged residues compared to Kin3 (described in more detail below) and, perhaps as a consequence, the C-terminus of Kin1 helix-α6 is connected by less density to the MT surface compared to Kin3 (*Figure 3B,D*, arrowhead). Thus, relatively limited conformational changes appear to accompany Mg-ADP release in the vicinity of helix-α6 and the neck linker. This is despite the previously described significant rearrangement of the switch loops at the nucleotide-binding site on the other side of the domain (*Figure 2*).

However on Mg-ATP binding, a major conformational change of helix-α6 is observed in both motors (*Figure 3C,E*; *Figure 3—figure supplement 1*). Compared to the NN reconstructions, helix-α6 and β-sheet1$_{abc}$ have together lifted and rotated away from the MT surface. In the Mg-ATP-like reconstructions, a hydrophobic cavity forms above helix-α4 (*Kikkawa et al., 2001*) because the central β-sheet has peeled away from its C-terminal end (see *Figure 3C,E*; and *Figure 3—figure supplements 2 and 3C,D,F,G*), helix-α6's C-terminus extends by a turn and inserts into this cavity. In the Kin3-Mg-ATP-like reconstruction, as a result of the repositioning of helix-α6, only a narrow bridge of density connects its N-terminal end with α–tubulin (*Figure 3C*, arrowhead). This N-terminal end is more negatively charged than the C-terminal end of helix-α6 that was in contact with the MT surface prior to Mg-ATP binding. In Kin1, density for helix-α6 disconnects from the MT surface altogether (*Figure 3E*, arrowhead). Importantly, in both motors, this structural reorganisation allows the neck linker to extend towards the MT plus end and dock along strand-β8 of the central β-sheet (*Figure 3C,E* and *Figure 3—figure supplement 3C,D,F,G*) (*Rice et al., 1999*). The N-termini of both motors are also directed towards the MT plus end, lying across the docked neck linker to form the CNB (*Figure 3—figure supplement 3C,D,F,G* and *Figure 4C,E*) (*Hwang et al., 2008*; *Khalil et al., 2008*). Thus, concerted conformational changes involving a number of structural elements appear to contribute to movement of helix-α6 and neck linker docking.

## A stable motor domain–MT interface is maintained through the ATPase cycle

These analyses show that in both Kin1 and Kin3, the same, small conformational changes at the nucleotide binding site on Mg-ATP binding have large structural consequences elsewhere. One important aspect of transmission of this mechanochemical information is that a stable interaction with the MT is

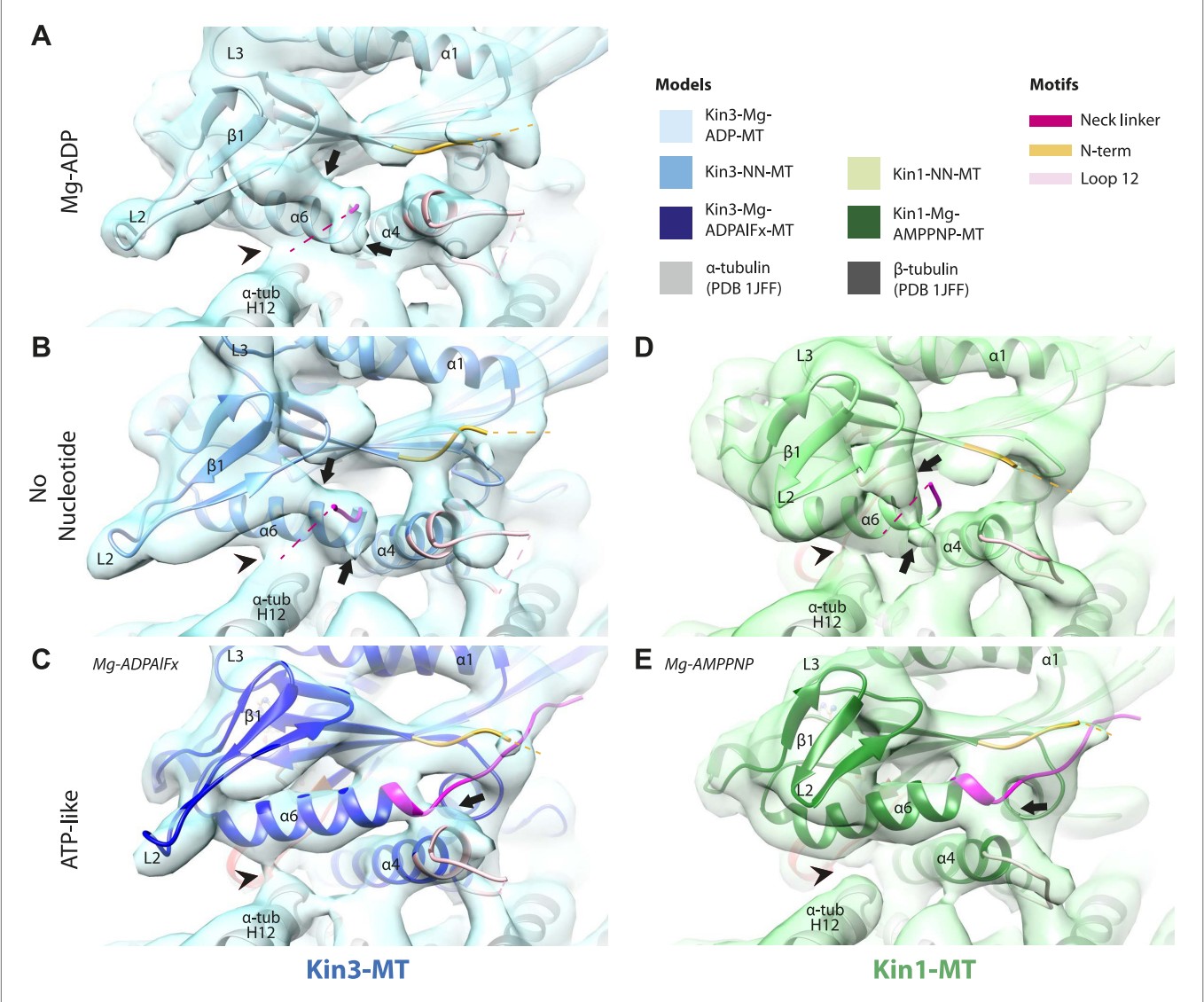

**Figure 3**. Conserved conformational changes of helix-α6 alter MT connectivity and allow neck linker docking on Mg-ATP binding. (**A–C**) View of helix-α6 and the neck linker (in fuchsia) of MT-bound Kin3 reconstructions (shown as blue transparent density) in (**A**) Mg-ADP, model shown in light blue, (**B**) no nucleotide (NN), model shown in mid-blue, (**C**) Mg-ADPAlFx, model shown in dark blue; (**D–E**) View of helix-α6 and the neck linker (in fuchsia) of MT-bound Kin1 reconstructions (shown as green transparent density) in (**D**) no nucleotide, model shown in light green, (**E**) Mg-AMPPNP, model shown in dark green. In Mg-ADP (Kin3) and NN states (both motors), helix-α6 contacts the surface of α-tubulin (arrowhead) and its orientation with respect to helix-α4 ensures that the neck linker cannot dock. Regions of density at the C-terminal end of helix-α6 likely representing conformers of the N-terminal portion of the neck linker are observed (arrows), although the majority is not visible, presumably due to flexibility. In both motors, peeling of the motor domain β-sheet core away from helix-α4 upon Mg-ATP binding allows rotation and extension of helix-α6, drawing it away from the MT surface (arrowhead), and allowing it to occupy the space between helix-α4 and the β-sheet core. The neck linker docks towards the MT plus end (arrow) and forms the CNB with the N-terminus (in orange). In all reconstructions, density for the motor domain was contoured to an equivalent volume.

The following figure supplements are available for figure 3:

**Figure supplement 1**. Conserved conformation of helix-α6 allows neck linker docking on Mg-ATP binding in Kin3 and Kin1 alternative ATP-like states.

**Figure supplement 2**. Tilting of the core β-sheet on Mg-ATP binding in Kin1 and Kin3 causes peeling of the β-sheet from the C-terminus of helix-α4 to allow movement and extension of helix-α6 and neck linker docking.

**Figure supplement 3**. Conserved conformational changes of helix-α6 relative to helix-α4 control neck-linker docking along the core β-sheet when Mg-ATP binds.

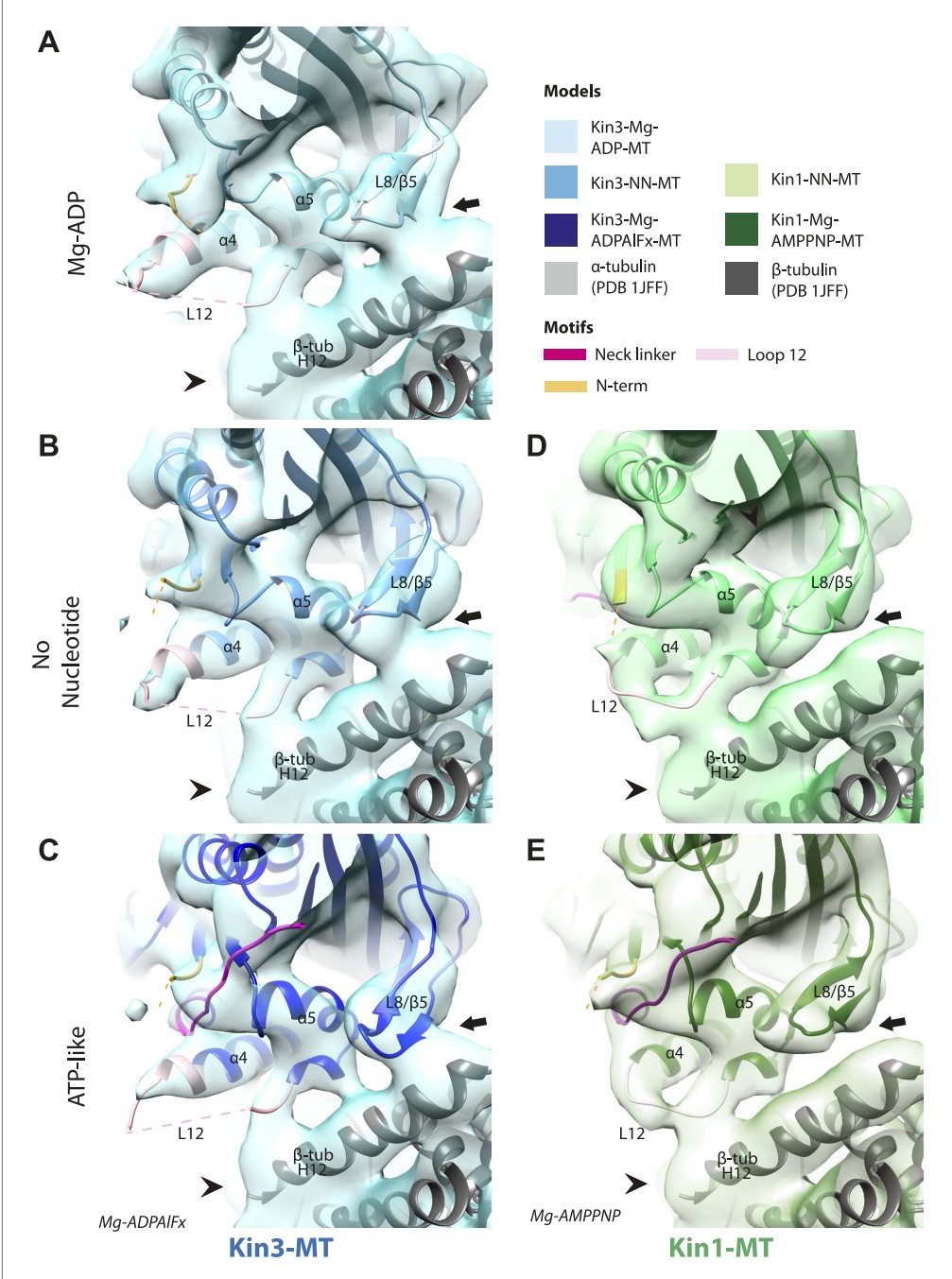

**Figure 4**. Nucleotide-independent interactions between the kinesin motor domain and the MT surface. (**A**–**C**) View from the MT plus end of the motor domain-MT interface in MT-bound Kin3 reconstructions (shown as blue transparent density) in (**A**) Mg-ADP, model shown in light blue, (**B**) no nucleotide (NN), model shown in mid-blue, (**C**) Mg-ADPAlFx, model shown in dark blue, in which the CNB is formed between the neck linker (fuchsia) and N-terminus (orange). The N-terminus of loop12 (light pink) extends helix-α4 by a turn but the central, lysine-rich portion of this loop is not visible (dotted pink line), nor is the β-tubulin CTT (arrowhead) with which it is known to interact. Loop8/strand-β5 forms a clear connection to the MT surface (arrow). (**D**–**E**) The same view of the motor domain-MT interface in MT-bound Kin1 reconstructions (shown as green transparent density) in (**D**) no nucleotide, model shown in light green, (**E**) Mg-AMPPNP, model shown in dark green, in which the CNB is formed between the neck linker (fuchsia) and N-terminus (orange). The shorter Kin1 loop12 is clearly visualised and contacts the MT
*Figure 4. Continued on next page*

*Figure 4. Continued*
surface while loop8/strand-β5 are not connected by density to the MT surface (arrow). In all reconstructions, density for the motor domain was contoured to an equivalent volume.
The following figure supplement is available for figure 4:

**Figure supplement 1**. Conserved conformations at the kinesin motor domain and the MT surface in Kin3 and Kin1 alternative ATP-like states.

sustained. Our data show that several structural elements form apparently invariant contacts with the MT (primarily β-tubulin) in all the nucleotide states we examined. In the Kin3 reconstructions, density corresponding to helix-α4 runs across the whole motor domain–MT interface (*Figure 4A–C*). At its C-terminal end, density corresponding to the N-terminal portion of the extended Kin3 loop12 sequence is stabilised as a helical turn (*Figure 4A–C*, pink). However, density corresponding to the middle Kin3-characteristic Lys-rich portion of this loop (the so-called K-loop) is not visible in any nucleotide state (*Figure 4A–C*, pink dotted line). This suggests that this highly basic middle section of loop12 remains mobile even while close to the MT surface (discussed below). The C-terminal end of Kin3 loop12, on the other hand, is visible and is stabilised by interaction with β-tubulin. Loop12 leads into an interconnected region of contacts between the MT surface and the motor, composed of helix-α5 along with loop8/strand-β5. These elements do not alter their interaction with the MT in the different nucleotide states calculated (*Figure 4A–C*; *Figure 4—figure supplement 1*).

The Kin1 reconstructions show the same structural components at the motor domain–MT interface, which are also invariant in the different nucleotide states (*Figure 4D,E*). In the Kin1 reconstructions—as with Kin3—helix-α4 forms a major contact at the tubulin intradimer interface and adopts a conserved orientation relative to the MT (*Figure 4D,E*). However, the C-terminus of the Kin1 helix-α4 is shorter by one turn compared to Kin3 because its loop12 is shorter and also lacks the lysine cluster characteristic of Kin3s (compare e.g. *Figure 4B,D*). Density corresponding to the Kin1 loop12 connects directly to helix-α5 at the MT interface (*Figure 4D,E*; *Figure 4—figure supplement 1*). However, in contrast to Kin3, there is no density in our reconstructions connecting Kin1 loop8/strand-β5 and the MT surface (*Figure 4D,E*).

## Mechanical amplification and force generation involves conformational changes across the motor domain

A key conformational change in the motor domain following Mg-ATP binding is peeling of the central β-sheet from the C-terminus of helix-α4 increasing their separation (*Figure 3—figure supplement 2*); this is required to accommodate rotation of helix-α6 and consequent neck linker docking (*Figure 3B–E*). Peeling of the central β-sheet has previously been proposed to arise from tilting of the entire motor domain relative to static MT contacts, pivoting around helix-α4 (the so-called 'seesaw' model; *Sindelar, 2011*). Specifically, this model predicts that the major difference in the motor before and after Mg-ATP binding would be the orientation of the motor domain with respect to helix-α4 (*Vale and Milligan, 2000*). Globally, the conformations of both Kin1 and Kin3 in our reconstructions are consistent with motor domain tilting of 12–15° on Mg-ATP binding (*Figure 3B–E*, *Figure 3—figure supplement 2*). In both motors, subtle flexure of the central β-sheet itself is also apparent on Mg-ATP binding (*Figure 5—figure supplement 1*) such that loop7 and the bottom of strand-β3 that connects to the P-loop are not superimposable. Differences in the β-sheet when comparing the Kin3-Mg-ADP and Kin3-NN models are even smaller in comparison (*Figure 5—figure supplement 1A*). In myosin, the equivalent structural region undergoes substantial β-sheet flexure on nucleotide release (backbone RMSD > 3.2 Å, *Figure 5—figure supplement 1D*; *Coureux et al., 2003*; *Reubold et al., 2003*). However, our data provide no evidence of significant flexing in the kinesin β-sheet that has been proposed to accompany Mg-ADP release (*Kull and Endow, 2013*). Furthermore, although the slight β-sheet bending that occurs when Mg-ATP binds may contribute to force generation as previously suggested (*Gigant et al., 2013*), it cannot, by itself, account for the peeling of the β-sheet that allows neck linker docking.

If motor domain tilt was sufficient to account for the mechanochemical transmission that takes place on Mg-ATP binding, superposition of the β-sheets of the NN and Mg-ATP structural states would be predicted to bring the motor domains into alignment (apart from helix-α4 and the nucleotide-invariant MT contacts). However, such a superposition shows large residual differences in multiple regions of the motor domain (*Figure 5A,B*; depicted as RMSDs between each pair of NN/Mg-ATP

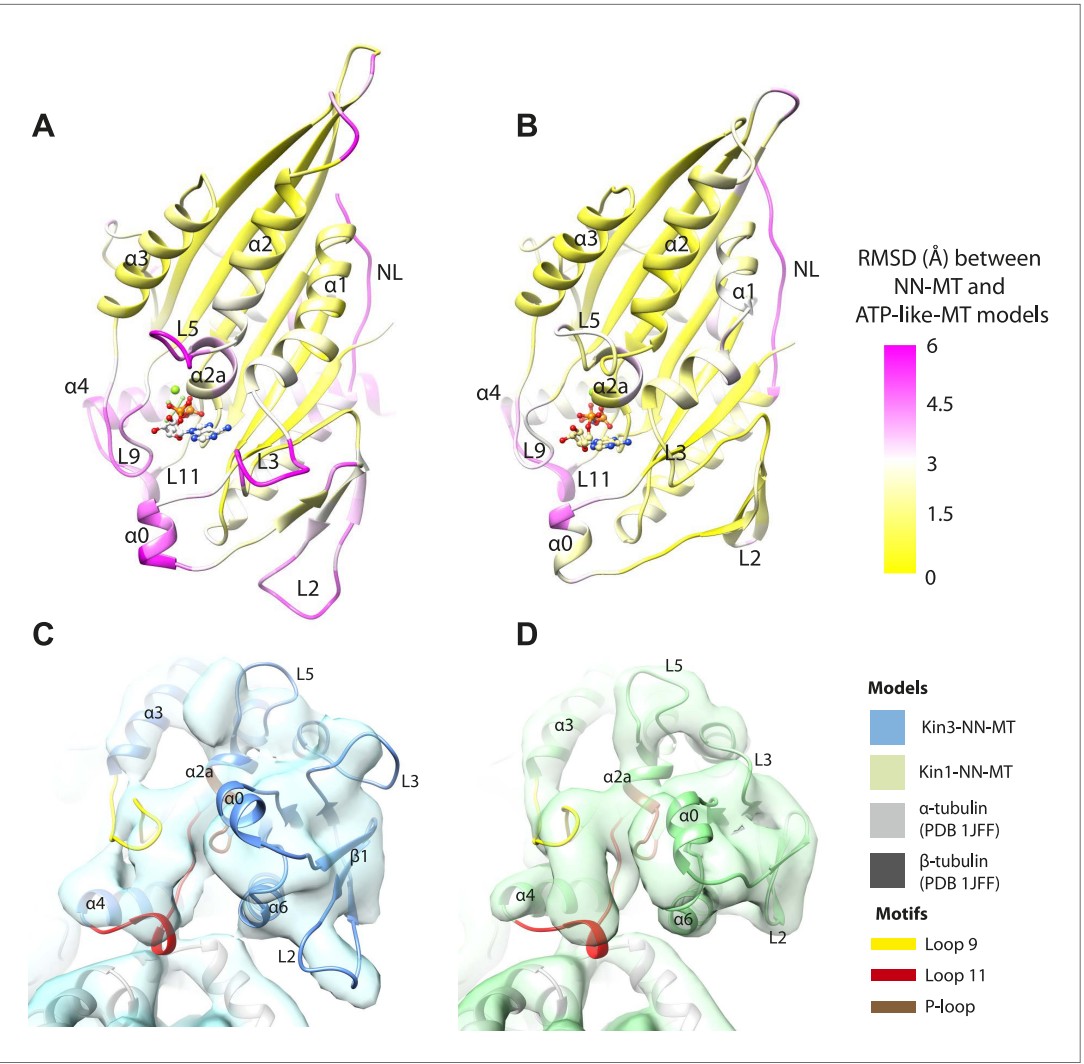

**Figure 5**. Transmission of force generation across the motor domain on Mg-ATP binding. (**A** and **B**) Conformational changes relative to superposition of the core β-sheet of Kin3 (**A**) and Kin1 (**B**) showing the RMSDs due to Mg-ATP binding coloured from yellow (no change) to pink (large change), depicted on the Mg-ATP-like structures. Note, because the core β-sheet moves relative to helix-α4, which is held at the MT interface, alignment of the β-sheet artificially shows large displacements of helix-α4 and other nucleotide-invariant MT contacts at the back of this view. (**C** and **D**) Comparison of the nucleotide-binding site before and after Mg-ATP binding in Kin3 (**C**) and Kin1 (**D**). In each case, the NN model is depicted within the Mg-ATP cryo-EM density and shows that the regions of the largest RMSDs (pink in panels **A** and **B**) correspond to regions of the models that clearly do not fit in the density, that is, that undergo conformational changes when Mg-ATP binds.

The following figure supplements are available for figure 5:

**Figure supplement 1**. Limited β-sheet flexure during kinesin ATPase cycle compared to myosin5.

**Figure supplement 2**. Pincer-like closure of loop9 and loop11 contributes to motor domain tilt when ATP binds.

models). This clearly demonstrates that the β-sheet tilting that occurs in the transition from NN to Mg-ATP is not sufficient to describe the conformational changes in either Kin3 or Kin1. This is further emphasized when the Kin3 and Kin1 NN pseudo-atomic models are superimposed on the β-sheets of their respective ATP-like docked models and compared to the Mg-ATP-like cryo-EM reconstructions (*Figure 5C,D*). Various parts of the NN models protrude from the density for the ATP-like reconstructions illustrating the poor fit, agreeing with the RMSD calculations, and further supporting their

tilt-independent movements (*Figure 5C,D* compare to *Figure 2C,E*). At the nucleotide-binding site, this analysis highlights that movement of loop9 around the bound Mg-ATP is large compared to motor domain tilting. Similarly, while loop11 retains a similar conformation before and after Mg-ATP binding, it does not tilt along with the core β-sheet but instead moves towards the motor domain core (see *Figure 5—figure supplement 2*). In addition, helix-α2a and loop5 above the nucleotide-binding site, and helix-α0 below the nucleotide-binding site, accommodate Mg-ATP binding in both motors (*Figure 5A,B*). Some structural changes are seen in helix-α1, whereas the β-sheet1$_{abc}$ shows clear conformational differences; family-specific loop insertions in loop2 and loop3 particularly exaggerate these movements in Kin3 (*Figure 5C*). The expected extension of helix-α6 and neck-linker docking is also highlighted by this analysis. However, it is also apparent that helix-α6 movement cannot be described purely by motor domain tilt, because it also undergoes a translational shift towards the MT plus end, as was recently proposed for Kin1 (*Gigant et al., 2013*). The improved resolution of our reconstructions thus allows us to conclude that the conformational changes that underlie force generation in both Kin1 and Kin3 involve: (1) motor domain tilting relative to static MT contacts, but also (2) more complex sets of movements that accommodate Mg-ATP binding and bring about mechanical amplification.

## Differences in the Kin1/Kin3 MT interface provide structural insight into superprocessivity of Kin3s

Despite high structural and mechanistic similarity between Kin3 and Kin1, contacts across the motor domain–MT interface are likely to contribute to differences in these motors' transport properties (*Figure 6*). One major difference is the presence of a Lys-rich insertion in Kin3 loop12 (the 'K-loop') (*Figure 6A*, pink shading) (*Okada and Hirokawa, 1999*). In Kin3s, loop12 mediates 1D diffusion of ADP-bound monomeric and dimeric Kin3s along MTs via flexible, electrostatic interactions with the acidic C-terminal tails (CTTs) of tubulin (*Okada and Hirokawa, 1999*, *2000*; *Kikkawa et al., 2000*; *Soppina et al., 2014*). The K-loop also enhances the initial interaction between Kin3 dimers and their track prior to processive stepping (*Soppina and Verhey, 2014*). In addition, whereas the catalytic turnover of Kin3 compared to Kin1 monomers are similar (our data in *Table 3* and e.g., *Okada and Hirokawa, 2000*), steady state ATPase assays show that the $K_m$MT of Kin3 is several hundred times lower than Kin1, a difference that depends partly on the K-loop (*Okada and Hirokawa, 2000*). Since the $K_m$MT is indicative of the MT affinity of ADP-bound kinesin (*Woehlke et al., 1997*), this is consistent with the role of the Kin3 loop12 in enhancing the association of Mg-ADP Kin3s with MTs (*Okada and Hirokawa, 1999*, *2000*; *Kikkawa et al., 2000*; *Soppina and Verhey, 2014*).

There is no density corresponding to the K-loop—nor of the tubulin CTTs with which it is proposed to interact—in any of our Kin3 reconstructions (*Figure 4A–C*). Given that density corresponding to Kin1 loop12 (*Figure 4D,E*) and Kin3 loops of equivalent size (e.g. loops 2 and 3 [7 and 8 residues respectively], *Figure 3A–C*) are clearly visualised, this suggests that this region of Kin3 is structurally heterogeneous, and therefore invisible in the context of our averaging methods. The K-loop may be intrinsically flexible due to its sequence, consistent with its role in mediating 1D diffusion. In addition, the lack of structural detail in this region could be due to the biochemical heterogeneity (different isoforms and post-translational modifications) of the CTTs of the bovine tubulin used in our experiments. Our structures imply that conformational flexibility of the K-loop persists throughout the motor's ATPase cycle but more information from future experiments is needed to clarify the contribution of this region to motor function.

However, the K-loop is reported to account for only a 10-fold enhancement of MT association of monomeric Kin3s over Kin1s (*Okada and Hirokawa, 1999*, *2000*), implying that other regions of the Kin3 motor domain also contribute. Our data show clear structural differences between Kin1 and Kin3 at the interface of the acidic tip of α-tubulin H12 with helix-α6, especially in the Mg-ADP/NN reconstructions (*Figure 3*). In addition, more subtle differences in the distribution of charged residues in loop11 and helix α4's N-terminus would be predicted to influence MT affinity (*Figure 6D*). Sequence divergence in loop8/strand-β5 was previously proposed to enable discrimination of post-translational modification in α-tubulin CTTs by Kin3 compared to Kin1 (*Konishi and Setou, 2009*). A direct role for recognition of the α-tubulin CTT is unlikely given its distance from loop8/strand-β5. However, differences in connectivity between this region of the motor domain and β-tubulin when comparing Kin1 and Kin3 (*Figure 4*) could contribute to differences in their apparent overall affinity. Intriguingly, recent data show that the K-loop does not contribute to the super-processive stepping properties of Kin3 dimers (*Soppina and Verhey, 2014*). Although a number of motor parameters could in principle

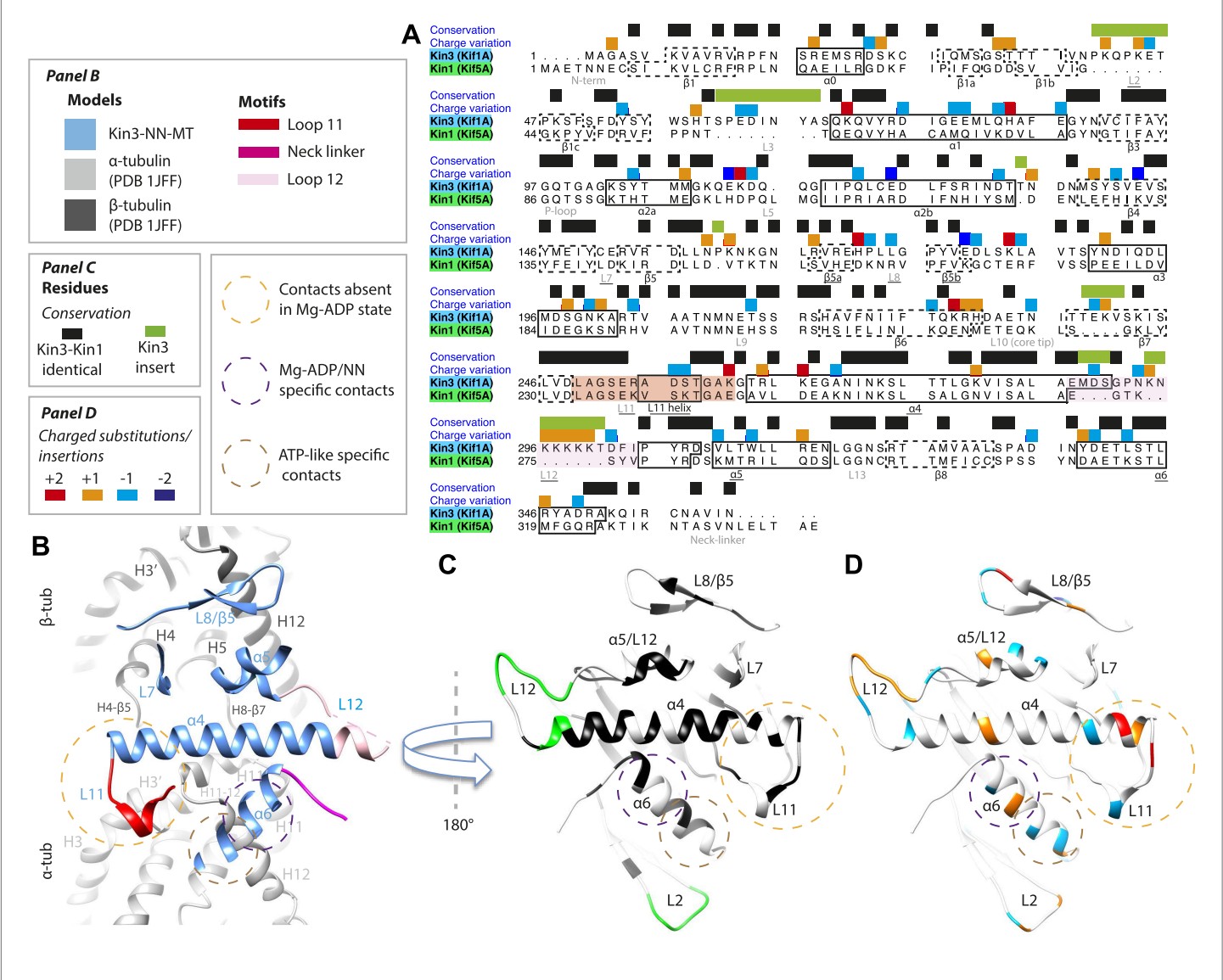

**Figure 6**. Comparison of Kin3 and Kin1. (**A**) Sequence alignment of Kin3 (Kif1A) and Kin1 (Kif5A) motor domains showing secondary structural elements within the domains, annotated according to sequence and charge conservation. Elements depicted in other panels are underlined. (**B**) Longitudinal slice through the Kin3-NN model viewed from the front showing the MT contact elements and the underlying structural regions in αβ-tubulin. (**C**) MT binding surface of Kin3-NN model viewed from the MT surface (180° rotated compared to **B**) annotated by sequence identity (black) between Kin3 and Kin1 and sequence insertions (green). Structural elements in the MT are removed in this view to most clearly show elements in the motor domain. (**D**) MT binding surface of Kin3-NN model showing the differences in charge (blue: Kin3 more acidic than Kin1; red: Kin3 more basic than Kin1); same view as in **C**.

contribute to processivity (e.g., coordination between dimer motor domains via the NL [*Clancy et al., 2011*]), our structures suggest that other regions of the Kin3–MT interface may also influence functional differentiation of these motors including super-processivity (*Figure 6C,D*).

## Discussion

Kinesin mechanochemistry and the extent of mechanistic conservation within the motor superfamily are open questions, critical to explain how MT binding, and ATP binding and hydrolysis drive motor activity. Our structural characterisation of two transport motors now allows us to propose a model that describes the roles of mechanochemical elements that together drive conserved MT-based motor function (*Figure 7*).

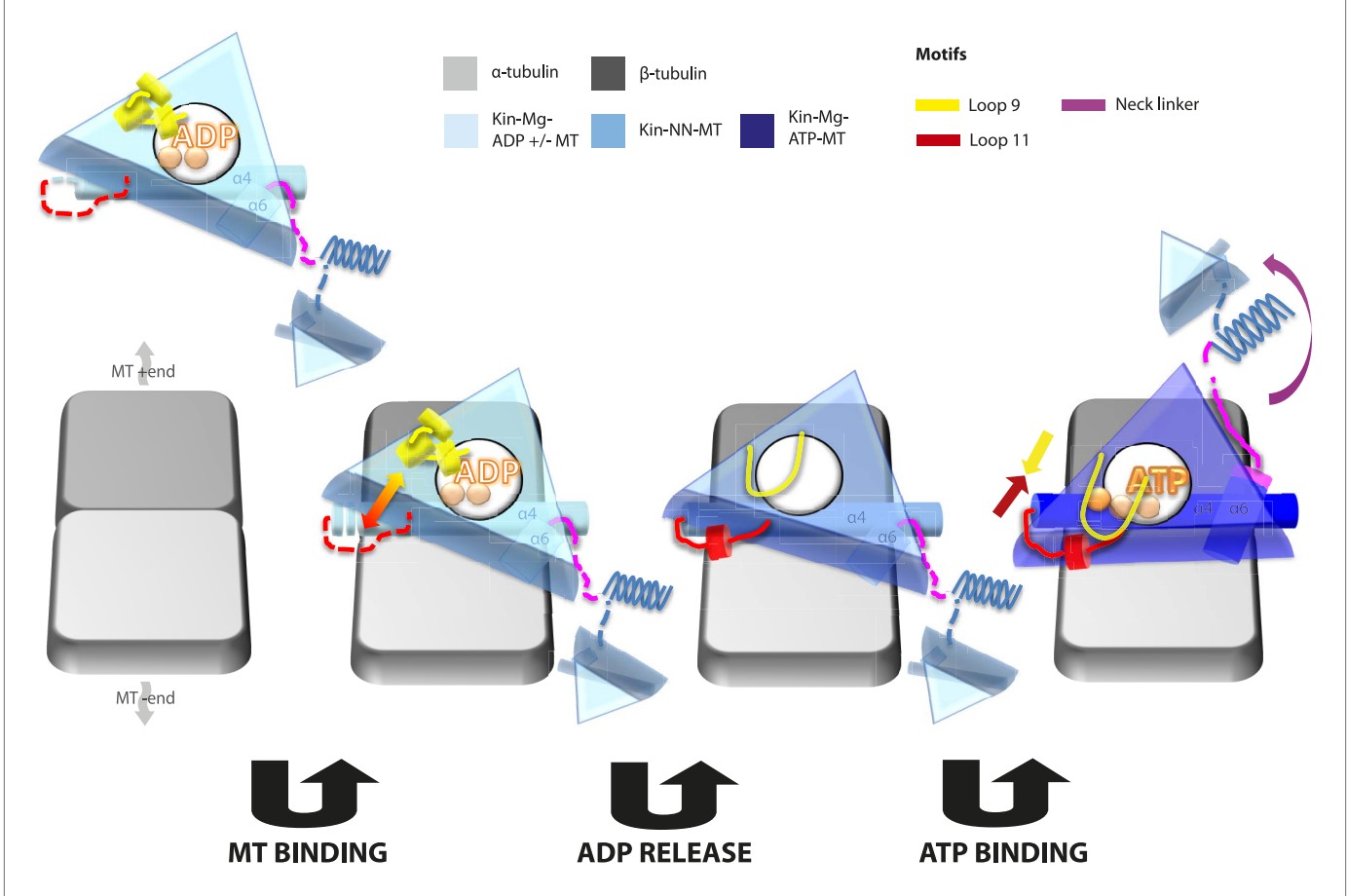

**Figure 7**. Model of conserved MT-bound kinesin mechanochemistry. Loop11/N-terminus of helix-α4 is flexible in ADP-bound kinesin in solution, the neck linker is also flexible while loop9 chelates ADP. MT binding is sensed by loop11/helix-α4 N-terminus, biasing them towards more ordered conformations. We propose that this favours crosstalk between loop11 and loop9, stimulating ADP release. In the NN conformation, both loop11 and loop9 are well ordered and primed to favour ATP binding, while helix-α6—which is required for mechanical amplification–is closely associated with the MT on the other side of the motor domain. ATP binding draws loop11 and loop9 closer together; causing (1) tilting of most of the motor domain not contacting the MT towards the nucleotide-binding site, (2) rotation, translation, and extension of helix-α6 which we propose contributes to force generation, and (3) allows neck linker docking and biases movement of the 2nd head towards the MT plus end.

In the Mg-ADP-bound kinesin, association with the MT surface is experienced directly by loop11 and the N-terminus of helix-α4 biasing their conformations towards more structured states. Full stabilisation of these elements is not achieved until Mg-ADP is released, and the additional contacts with the MT surface may in particular serve to nucleate the single turn helix in loop11. This is consistent with the well-documented role of loop11 in sensing MT attachment and triggering Mg-ADP release via interactions with α-tubulin (*Woehlke et al., 1997*; *Yun et al., 2001*; *Ebbing et al., 2008*; *Uchimura et al., 2010*). Loop9 does not directly contact the MT before or after Mg-ADP release but dramatically changes conformation, unfurling, and extending around the nucleotide-binding site. The structured conformations of loop11 and the N-terminus of helix-α4 are sterically compatible with the conformations of loop9 before and after Mg-ADP release—that is, no clashes are seen in either case. However, the extended conformation of loop9 and the ordered conformations of helix-α4/loop11 are likely to be mutually stabilising due to formation of additional contacts, and thereby mediate communication between the nucleotide and MT-binding sites (*Woehlke et al., 1997*; *Yun et al., 2001*; *Farrell et al., 2002*; *Ebbing et al., 2008*; *Nitta et al., 2008*). Critically, however, the water network coordinating Mg-ADP is stabilized exclusively by the retracted helical conformation of loop9 (*Figure 2—figure supplement 2*). The transition towards the extended conformation of loop9 promotes Mg-ADP release by destabilisation of Mg coordination (*Nitta et al., 2008*). These structural rearrangements therefore

indicate that sequential conformational changes of the switch loops in the presence of MTs stimulate Mg-ADP release, the rate-limiting step of motors in solution (*Hackney, 1988*). These rearrangements allow formation of a nucleotide-free motor that is strongly bound to its MT track (*Nakata and Hirokawa, 1995*), at least in part due to additional contacts formed between loop11 and the MT.

Conformational changes at the nucleotide-binding site that lead to Mg-ADP release also appear to prime the kinesin motor domain for Mg-ATP binding. However, the primed conformation clearly does not lead to neck linker docking in the absence of Mg-ATP, contrary to previous predictions (*Nitta et al., 2008*). Multiple strands of evidence suggest that the neck linkers of transport kinesins in solution explore both docked and undocked conformations independent of the nucleotide state (*Rice et al., 1999*; *Nitta et al., 2008*; *Scarabelli and Grant, 2013*). Thus, tight MT binding is critical in strongly biasing neck linker conformation in the absence of nucleotide such that it will be undocked and, in our reconstructions, directed albeit flexibly towards the MT minus end. Interaction of helix-α6 with α-tubulin's H12 (*Uchimura et al., 2010*) may therefore help to prevent neck linker docking in the absence of nucleotide, despite changes in the conformations of the switch loops at the active site.

Mg-ATP binding does not cause large rearrangements of the nucleotide-binding site of MT-bound motor domains. However, the presence of the pre-hydrolysis γ-phosphate of Mg-ATP is critical for the pincer-like movement of loop11 and loop9 towards each other. Along with formation of strong additional contacts between these loops, the helix-α4 N-terminus and the P-loop (see *Figure 2—figure supplement 4* and *Parke et al., 2010*; *Chang et al., 2013*; *Gigant et al., 2013*), this new local connectivity induces the larger rearrangements that cause neck linker docking. The resulting conformational changes cannot be described only as a tilt of the motor domain relative to static contacts with the MT including helix-α4; in addition to β-sheet tilting, multiple changes across the domain reinforce mechanical amplification and neck linker docking when Mg-ATP binds. The resolution of our reconstructions also allows us to detect subtle distortion of the central β-sheet edges on Mg-ATP binding. However, arguably the most important consequences of Mg-ATP binding are the changes—extension, tilting, and translation—in helix-α6 that allow neck linker docking. This conformation is stabilised by contacts between its N-terminus and elements in the nucleotide-binding pocket (see *Figure 2—figure supplement 4* and *Parke et al., 2010*; *Chang et al., 2013*; *Gigant et al., 2013*).

Neck linker docking is essential for both defining the directionality of kinesin motility and mediating head–head tension to ensure processive dimer stepping (*Rice et al., 1999*; *Tomishige and Vale, 2000*; *Vale and Milligan, 2000*; *Skiniotis et al., 2003*; *Clancy et al., 2011*; *Sindelar, 2011*), but whether docking itself can generate the force required for kinesin stepping has been questioned (*Rice et al., 2003*). Thus, the structural basis of ATP-dependent force generation remains a matter of debate in the field (*Visscher et al., 1999*; *Cross and McAinsh, 2014*). The conformational changes associated with helix-α6 during the ATPase cycle—in which contacts with the MT formed in the ADP/NN state are broken as Mg-ATP-dependent rotation pulls it away from the MT surface—reinforce neck linker movements and may also contribute to mechanical amplification and force generation. The translation/extension of helix-α6 into the hydrophobic cavity that is created by β-sheet tilting when Mg-ATP binds may ensure that this tilting is not reversed. Intriguingly, mutagenesis of residues at the helix-α6/neck linker junction has a profound effect on the activity of kinesin monomers (*Case et al., 2000*), pointing to the importance and likely conservation of structural transitions in this region (*Case et al., 1997*). Importantly, movement of helix-α6 also relieves steric blocking of neck linker docking and presumably biases the mobile neck linker trajectory. In collaboration with the motor N-terminus, formation of the CNB reinforces the plus end directionality of this bias. Thus, we propose that the helix-α6 is a key mechanical element within the kinesin motor domain, and that its Mg-ATP-dependent movement is essential to plus-end directed stepping.

Once the neck linker has docked, ATP hydrolysis occurs, ensuring efficient coupling between kinesin stepping, Mg-ATP binding and hydrolysis (*Schnitzer et al., 2000*; *Hahlen et al., 2006*). A detailed reaction mechanism for hydrolysis has been proposed based on the conformations of loop9 and loop11 (a so-called 'phosphate tube') with Mg-ATP-analogue bound (*Parke et al., 2010*). Consistent with MT binding being important in the catalytic enhancement of kinesins (*Ma and Taylor, 1997*), this hydrolysis competent configuration of the switch loops is rarely seen in Mg-ATP-analogue kinesin structures in the absence of MTs (e.g., *Kikkawa et al., 2001*; *Nitta et al., 2004*; *Cochran et al., 2009*, with *Parke et al., 2010*; *Chang et al., 2013* being the notable exceptions); those in complex with tubulin always adopt this configuration (*Sindelar and Downing, 2010*; *Goulet et al., 2012*; *Gigant et al., 2013*). On Mg-ADP release, loop9 and loop11 are stabilized into conformations quite close to

catalytically competent ones. This suggests that the conformational changes triggered by MT binding that lead to MT-stimulated ADP release also contribute to setting up the catalytic site for ATP hydrolysis. Thus, a subset of mutations in MT-sensing residues in loop11 or which decouple MT affinity and ADP-release also affect MT-stimulated ATP-hydrolysis (*Woehlke et al., 1997*; *Song and Endow, 1998*; *Yun et al., 2001*; *Ebbing et al., 2008*; *Uchimura et al., 2010*). Following hydrolysis and phosphate release, we would predict that the Mg-ADP remaining in the catalytic site causes retraction of loop9, subsequent destabilization of loop11 and the helix-α4 N-terminus, leading to track detachment.

This model allows several previously proposed hypotheses, in particular concerning MT-stimulated Mg-ADP release, to be excluded. Mechanisms that involve MT-induced 'opening' of the nucleotide pocket, disordering of the switch loops around the nucleotide pocket to destabilise Mg-ADP coordination, or in which loop9 extends into the nucleotide pocket to perturb the P-loop and eject Mg-ADP (*Yun et al., 2001*; *Kikkawa and Hirokawa, 2006*; *Sindelar and Downing, 2007*; *Nitta et al., 2008*; *Sindelar, 2011*) are not supported by our observations that: (1) both loop9 and loop11 move towards the nucleotide-binding pocket on Mg-ADP release, (2) these loops adopt well-defined and conserved conformations that are clearly visualised after Mg-ADP release, and (3) the conformation of these loops does not sterically interfere with nucleotide binding or disrupt the P-loop. Another prominent idea is that a significant twist of the core β-sheet caused by MT attachment would promote Mg-ADP release analogous to the equivalent release step in myosin (*Coureux et al., 2003*; *Hirose et al., 2006*; *Kull and Endow, 2013*). However, comparison of our Kin3-Mg-ADP and Kin3-NN reconstructions (*Figure 5—figure supplement 1A*) does not support β-sheet twist as a mechanism for Mg-ADP release in kinesins.

The structural elements involved in these mechanochemical transitions are extremely well conserved amongst kinesins, and it is likely that the mechanisms we describe are utilised by all superfamily members. We previously characterised the MT-bound ATPase cycle of human kinesin-5 (Kin5, *Goulet et al., 2012*, *2014*). Although the resolutions of those cryo-EM reconstructions (~10 Å) do not provide the level of detail of the current work, many of our current hypotheses are consistent with a conserved mechanochemistry, specifically conformational coupling of loops9 and 11 to bring about MT-induced Mg-ADP release and Mg-ATP induced neck linker docking. Superimposed on this conserved mechanochemistry, family-specific modifications were also detected; most strikingly for Kin5, these include the proposed role of the Kin5-extended loop5 in controlling nucleotide binding and the stiffer properties of the Kin5 neck linker that undergoes an order-to-order transition on Mg-ATP binding. Family-specific insertions elsewhere in the motor domain are likely to have other modifying roles, such as Kin3's loop12, which enhances the initial interaction between these highly processive motors and their tracks (*Soppina and Verhey, 2014*). A tantalising hint of how insertions in loop2 may be coupled to MT depolymerisation in for example kinesin-13s (*Desai et al., 1999*; *Moores et al., 2002*; *Asenjo et al., 2013*) and kinesin-8s (*Varga et al., 2006*; *Peters et al., 2010*) is provided by its proximity to the MT surface and the mechanical amplifier helix-α6, and by its large displacement on Mg-ATP binding. Future studies at high resolution will provide further insights into the ways this conserved mechanochemistry is modified in diverse functional contexts within the kinesin superfamily.

## Materials and methods

### Protein purification

A human kinesin-1 (Kin1) construct (Kif5A, residues 1–340, in pET151-D-TOPO [Invitrogen, Carlsbad, CA with a TEV protease-cleavable N-terminal His$_6$-tag]) was expressed recombinantly in *Escherichia coli* and purified using cobalt affinity chromatography. The His$_6$-tag was removed by cleavage with TEV protease, and the untagged protein was buffer exchanged into BrB20 buffer (20 mM PIPES, 2 mM MgCl$_2$, 1 mM EGTA, 2 mM DTT, pH 6.8). A human kinesin-3 (Kin3) construct (Kif1A, residues 1–361, in pFN18a (with a TEV protease-cleavable N-terminal Halo-tag and a C-terminal His$_6$-tag [a kind gift from Prof Christopher A Walsh's laboratory, Harvard Medical School]) was expressed recombinantly in *E. coli* and purified using nickel affinity chromatography and size exclusion chromatography (GE Healthcare Life Science, UK, Superdex 75). The N-terminal Halo-tag was removed by cleavage with TEV protease, the sample was dialyzed into storage buffer (20 mM HEPES, pH 7, 150 mM NaCl, 1 mM TCEP, 5 mM MgCl$_2$, and 0.1 mM ADP) and concentrated. Note that this construct contains the native Kin3 (Kif1A) sequence, as opposed to several previous studies where a chimeric protein with substitution of its neck linker with that of the kinesin-1 Kif5C (*Kikkawa et al., 2001*; *Nitta et al., 2004*;

*Kikkawa and Hirokawa, 2006*; *Nitta et al., 2008*). The steady-state MT-activated ATPase activities of our motor constructs were determined by measuring phosphate production with a commercially available kit (EnzChek, Molecular Probes, Eugene, OR). Assays contained 10 nM motor domain and a minimum of fourfold molar excess of paclitaxel-stabilised MTs in 50 mM K-acetate, 25 mM HEPES, 5 mM Mg-acetate, 1 mM EGTA, pH 7.5 at 20°C. The dependence of rates of inorganic phosphate production on [MT] and [ATP] was fitted with a Michaelis–Menten relationship (*Table 3*).

## Microtubule preparation

Bovine tubulin (Cytoskeleton Inc, Denver, CO) at a final concentration of 50 µM in MT polymerization buffer (100 mM MES, pH 6.5, 1 mM $MgCl_2$, 1 mM EGTA, 1 mM DTT, 5 mM GTP) was polymerized at 37°C for 1 hr. 1 mM paclitaxel (Calbiochem, San Diego, CA) in DMSO was then added, and the sample was incubated at 37°C for a further hour.

## Cryo-EM sample preparation

MTs were diluted in BrB20 to a final concentration of 5 µM. Kin1 and Kin3 were diluted in BrB20 containing either 2 mM of AMPPNP, ADP, ADP + $AlF_4$, or apyrase (10 units/ml), according to established protocols (*Hirose and Amos, 2007*; *Sindelar and Downing, 2007*, *2010*; *Fourniol and Moores, 2011*), and warmed to room temperature 10 min prior to complex formation. The final concentrations used to visually achieve full decoration in the various nucleotide states are shown in *Table 4*. C-flat holey carbon grids (Protochips, Raleigh, NC) with 2 µm holes and 4 µm spacing were glow-discharged in air. 4 µl drops of MT then Kin1 or Kin3 samples were added and blotted in sequential fashion using a Vitrobot plunge-freezing device (FEI Co., Hillsboro, OR) operating at 25°C and 100% humidity and vitrified in liquid ethane.

## Data collection

Images of MT-kinesin complexes were collected using a 4k × 4k CCD camera (Gatan Inc., Pleasanton, CA) on a FEI Tecnai G2 Polara operating at 300 kV with a calibrated magnification of 100,000× and a final sampling of 1.5 Å/pixel. A defocus range of 0.4–3.5 µm and an electron dose of ~20 $e^-/Å^2$ were used. Images were screened manually to remove those with drift and/or objective astigmatism, contamination, and not containing at least one fully decorated and straight 13 protofilament MT.

## Data processing

Kinesin-decorated straight 13 protofilament MT segments were manually boxed using Eman suite's Boxer (*Ludtke et al., 1999*) and input to a set of custom-designed semi-automated single-particle processing scripts using Spider (*Frank et al., 1996*) and Frealign (*Grigorieff, 2007*) as described previously (*Sindelar and Downing, 2007*, *2010*), with minor modifications during local refinement. The phi-angle and thus seam location is determined in pseudo-symmetrical 13 protofilament MTs using projection matching in Spider (*Frank et al., 1996*). Once approximate alignment parameters are determined and manually verified (based on known values for the MT lattice), local refinement and CTF correction is performed in Frealign (*Grigorieff, 2007*). Eight rounds of refinement were undertaken and a negative B-factor of −400 was applied to the output reconstruction of round five to escape local minima in the search space; no B-factor was applied in the following three rounds to reduce possible over-fitting (http://grigoriefflab.janelia.org/forum). The angular distribution was isotropic for all data sets and the final reconstructions of the asymmetric unit (αβ-tubulin heterodimer + kinesin motor domain) were generated using 13 protofilament MT pseudo-symmetry. All final maps were assessed for possible over-fitting during refinement using a high-resolution noise-substitution test (*Chen et al., 2013*). Final estimated resolutions for each reconstruction are reported in *Table 1* and FSC curves are shown in *Figure 1—figure supplement 1*. Band-pass filtering of these reconstructions using

**Table 4.** Final protein concentrations used for cryo-EM sample preparation

| Kinesin and nucleotide state | [MT] (µM) | [Motor domain] (µM) |
|---|---|---|
| Kin3 MgADP | 5 | 10 |
| Kin3 NN | 5 | 5 |
| Kin3 Mg-AMPPNP | 5 | 5 |
| Kin3 Mg-ADP.AlFx | 5 | 5 |
| Kin1 NN | 5 | 100 |
| Kin1 Mg-AMPPNP | 5 | 50 |
| Kin1 Mg-ADP.AlFx | 5 | 50 |

Kin1 samples required higher concentrations than Kin3 to achieve good MT occupancy.

a Fermi temperature of 0.04 was performed in Spider (*Frank et al., 1996*) between frequencies of 15–6 Å (except for K1 Mg-ADPAlFx-MT reconstruction, where 15–7 Å was used).

## Atomic structure fitting and refinement

50 initial atomic models of each motor domain (in each nucleotide state) were built using Modeller v9.12 (*Sali and Blundell, 1993*) based on multiple template structures (see *Table 2*). Initial fitting of each model into the respective maps was done using the Chimera *fit_in_map* tool (*Goddard et al., 2007*). The best model was selected based on a combination of the cross correlation coefficient (CCC) between each model and the density map and a statistical potentials score (zDOPE; *Shen and Sali, 2006*). Each map was box-segmented around the motor domain, and the EM density for the tubulin was masked out (using Chimera *volume eraser* tool). The best fits were further refined with Flex-EM following a multistep optimisation protocol relying on simulated annealing molecular dynamics and a conjugate-gradients minimization applied to a series of subdivisions of the structure into rigid bodies (*Topf et al., 2008*) as identified by RIBFIND (*Table 2*; *Pandurangan and Topf, 2012*). In order to analyse subtle conformational changes occurring in various regions of the domain in the different nucleotide states, the quality of the final fits was assessed locally with TEMPy (Farabella et al., Unpublished) using the segment based cross-correlation coefficient (SCCC, *Figure 1—figure supplement 2*) (*Pandurangan et al., 2014*).

## Acknowledgements

The authors thank Charles Sindelar (Yale University, USA) for reconstruction algorithms, members of the Birkbeck EM group for helpful discussions, the MRC (MR/J000973/1; to JA, CAM), NIGMS (RO1GM102875-07; to SR), BBSRC (BB/K01692X/1; to IF, MT), and CNRS, la Ligue contre le Cancer Comité de Paris, ARC SFI20121205398 and the Fédération pour la Recherche sur le Cerveau (FRC; AH and I-MY) for funding. I-MY is a recipient of a Marie Curie IIF Fellowship. The team of AH is part of Labex Deep: 11-LBX-0044.

## Additional information

### Funding

| Funder | Grant reference number | Author |
|---|---|---|
| Medical Research Council | MR/J000973/1 | Joseph Atherton, Carolyn A Moores |
| National Institute of General Medical Sciences | RO1GM102875-07 | Steven S Rosenfeld |
| Biotechnology and Biological Sciences Research Council | BB/K01692X/1 | Irene Farabella, Maya Topf |
| Centre National de la Recherche Scientifique | | I-Mei Yu, Anne Houdusse |
| Ligue Contre le Cancer | | I-Mei Yu, Anne Houdusse |
| Federation pour la Recherche sur le Cerveau | | I-Mei Yu, Anne Houdusse |
| Marie Curie Fellowship | International Incoming Fellowship | I-Mei Yu |

The funders had no role in study design, data collection and interpretation, or the decision to submit the work for publication.

### Author contributions

JA, Conception and design, Acquisition of data, Analysis and interpretation of data, Drafting or revising the article; IF, AH, MT, Analysis and interpretation of data, Drafting or revising the article; I-MY, ; SSR, Acquisition of data, Analysis and interpretation of data, Drafting or revising the article; CAM, Conception and design, Analysis and interpretation of data, Drafting or revising the article

# Additional files

## Major datasets

The following datasets were generated:

| Author(s) | Year | Dataset title | Dataset ID and/or URL | Database, license, and accessibility information |
| --- | --- | --- | --- | --- |
| Atherton J, Farabella I, Yu IM, Rosenfeld SS, Houdusse A, Topf M, Moores CA | 2014 | 13-protofilament microtubule-bound human kinesin-3 motor domain in absence of nucleotides | http://www.ebi.ac.uk/pdbe/entry/EMD-2765 | Publicly available at The Electron Microscopy Data Bank. |
| Atherton J, Farabella I, Yu IM, Rosenfeld SS, Houdusse A, Topf M, Moores CA | 2014 | 13-protofilament microtubule-bound human kinesin-3 motor domain in absence of nucleotides | http://www.pdb.org/pdb/explore/explore.do?structureId=4uxo | Publicly available at RCSB Protein Data Bank. |
| Atherton J, Farabella I, Yu IM, Rosenfeld SS, Houdusse A, Topf M, Moores CA | 2014 | 3-protofilament microtubule-bound human kinesin-3 motor domain with AMPPNP | http://www.ebi.ac.uk/pdbe/entry/EMD-2766 | Publicly available at The Electron Microscopy Data Bank. |
| Atherton J, Farabella I, Yu IM, Rosenfeld SS, Houdusse A, Topf M, Moores CA | 2014 | 3-protofilament microtubule-bound human kinesin-3 motor domain with AMPPNP | http://www.pdb.org/pdb/explore/explore.do?structureId=4uxp | Publicly available at RCSB Protein Data Bank. |
| Atherton J, Farabella I, Yu IM, Rosenfeld SS, Houdusse A, Topf M, Moores CA | 2014 | 13-protofilament microtubule-bound human kinesin-3 motor domain with ADPAlFx | http://www.ebi.ac.uk/pdbe/entry/EMD-2767 | Publicly available at The Electron Microscopy Data Bank. |
| Atherton J, Farabella I, Yu IM, Rosenfeld SS, Houdusse A, Topf M, Moores CA | 2014 | 13-protofilament microtubule-bound human kinesin-3 motor domain with ADPAlFx | http://www.pdb.org/pdb/explore/explore.do?structureId=4uxr | Publicly available at RCSB Protein Data Bank. |
| Atherton J, Farabella I, Yu IM, Rosenfeld SS, Houdusse A, Topf M, Moores CA | 2014 | 13-protofilament microtubule-bound human kinesin-3 motor domain with ADP | http://www.ebi.ac.uk/pdbe/entry/EMD-2768 | Publicly available at The Electron Microscopy Data Bank. |
| Atherton J, Farabella I, Yu IM, Rosenfeld SS, Houdusse A, Topf M, Moores CA | 2014 | 13-protofilament microtubule-bound human kinesin-3 motor domain with ADP | http://www.pdb.org/pdb/explore/explore.do?structureId=4uxs | Publicly available at RCSB Protein Data Bank. |
| Atherton J, Farabella I, Yu IM, Rosenfeld SS, Houdusse A, Topf M, Moores CA | 2014 | 13-protofilament microtubule-bound human kinesin-1 motor domain in absence of nucleotides | http://www.ebi.ac.uk/pdbe/entry/EMD-2769 | Publicly available at The Electron Microscopy Data Bank. |
| Atherton J, Farabella I, Yu IM, Rosenfeld SS, Houdusse A, Topf M, Moores CA | 2014 | 13-protofilament microtubule-bound human kinesin-1 motor domain in absence of nucleotides | http://www.pdb.org/pdb/explore/explore.do?structureId=4uxt | Publicly available at RCSB Protein Data Bank. |
| Atherton J, Farabella I, Yu IM, Rosenfeld SS, Houdusse A, Topf M, Moores CA | 2014 | 13-protofilament microtubule-bound human kinesin-1 motor domain with AMPPNP | http://www.ebi.ac.uk/pdbe/entry/EMD-2770 | Publicly available at The Electron Microscopy Data Bank. |
| Atherton J, Farabella I, Yu IM, Rosenfeld SS, Houdusse A, Topf M, Moores CA | 2014 | 13-protofilament microtubule-bound human kinesin-1 motor domain with AMPPNP | http://www.pdb.org/pdb/explore/explore.do?structureId=4uxy | Publicly available at RCSB Protein Data Bank. |
| Atherton J, Farabella I, Yu IM, Rosenfeld SS, Houdusse A, Topf M, Moores CA | 2014 | 13-protofilament microtubule-bound human kinesin-1 motor domain with ADPAlFx | http://www.ebi.ac.uk/pdbe/entry/EMD-2771 | Publicly available at The Electron Microscopy Data Bank. |
| Atherton J, Farabella I, Yu IM, Rosenfeld SS, Houdusse A, Topf M, Moores CA | 2014 | 13-protofilament microtubule-bound human kinesin-1 motor domain with ADPAlFx | http://www.pdb.org/pdb/explore/explore.do?structureId=4uy0 | Publicly available at RCSB Protein Data Bank. |

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
