## [Decision Letter]

Thank you for sending your work entitled “Conserved mechanisms of microtubule-stimulated ADP release, ATP binding, and force generation in transport kinesins” for consideration at *eLife*. Your article has been favorably evaluated by John Kuriyan (Senior editor), a member of our Board of Reviewing Editors, and 3 reviewers.

The reviewers are in agreement about the high technical quality of your study, and the interest of your conclusions. They all agree that the strong side of the manuscript is the convincing evidence for changes in the nucleotide pocket and associated movements of h4 and h6.

One of the reviewers pointed out that the weak side comprises parts dealing with the movements of the neck linker, and the role of charged interactions in KIF1A (K-loop) and after discussion; the other reviewers agree that you need to more convincingly support your hypothesis.

We would like to publish your manuscript. However, to convincingly support the hypotheses on the neck linker, we feel you have to address the following issues that are detailed below. There is potentially quite a lot of work here, and we are not asking you to spend more than a couple of months to answer them. If you have data to address these issues, you should include them. If not, some of the data can be quite easily obtained, or we would recommend toning down your conclusions, and being more careful with potential caveats.

Movement of h6 and neck linker:

In the Results section it is argued that 'the neck linker remains directed towards the MT minus end...' under no-nucleotide conditions (Figure 3). In the figure, however, it is hard see electron density supporting this statement. In fact, in none of the Figure 3, Figure 3—figure supplement 1) can electron densities be unambiguously detected, corresponding to the neck linker. Figure 3—figure supplement 1 might be an exception but the density extending from h6 could also be attributed to the N-terminal extension (yellow), as seemingly the case for KIF1A (Figure 3—figure supplement 1). The data basis for interpretations on the position of the neck linker is too weak to allow such far-reaching conclusions. The neck linker orientation might be determined using label such as [69]. ([69], The EMBO Journal Vol. 22 No. 7 pp. 1518-1528, 2003).

The conclusion that 'the beta-sheet tilting ... is not sufficient to describe the conformational changes that accompany force generation...' is an important key conclusion, substantiated by the thorough analysis of alterations in the core motor domain (i.e. without neck linker).

You write (in the Results section) '... no density corresponding to [K-loop and CTT]..., nor ... any effect of motor domain nucleotide state...', and conclude: '... hard to reconcile with ... a specific role of L12 ... in the Mg-ADP state.' This logic is incorrect. The method used (cryo EM reconstruction) does not allow detection of any differences but it cannot be concluded that there ARE no differences (that might be detected by other methods). There may be differences in L12/CTT in dependence on the nucleotide state, just invisible to cryo EM.

Furthermore, in the Results section you state '[the ADP-dependent Kin3 motor and track] interaction is not important (or does not exist)...' As argued above, the kinetic data in Table 3 clearly hints at a tight, possibly ionic interaction. Likewise, a '10-fold affinity enhancement cannot [account for] super-processivity' is incorrect. If both heads of a dimeric motor were uncoupled (which they are not, but for the sake of the argument this is irrelevant) and each had a microtubule off-rate of 1/100 s-1, the probability for the dimer to have both heads in the weakly bound state would be (1/100)^2 = 1/10000, which would explain 'super-processivity'. In general, 'super-processive' is a concept that cannot be tested or substantiated in a study with monomeric kinesin motor domain constructs. Speculations on why KIF1A dimers show long run lengths based on structural studies on monomers are highly hypothetical. At least, one would have to know whether 8-nm steps occur, and whether they are coupled 1:1 to ATP hydrolysis.

In the Results section you also state '...the apparent affinity of Kin3 is ∼250x higher than Kin1', and refers to Table 3. This table, however, reports catalytic parameters, not affinities.

This issue is important for the interpretation of the role of the K-loop in KIF1A. Co-sedimentation assays at varying ionic strengths can reveal the relative importance of ionic and other interactions, possibly in combination with mutants.

That KIF1A's observed half-maximal activation constant for microtubules (Table 3) is heavily affected by ionic or unspecific interactions can be seen by calculating the kcat/K0.5MT ∼800 s-1 uM-1, which is unrealistically high for a diffusion-limited reaction. The authors have to clearly distinguish between affinity and half-maximal activation constant.

What is the evidence for the actual presence of Mg-ADP in the nucleotide binding pocket under ADP conditions? Figure 2 shows KIF1A under this condition but due to the projection into 2D it is not visible whether there is a clear density signal corresponding to the indicated position of Mg-ADP. In this respect, Figure 2 is much less clear than Figure 2 (ADP/AlFx).

Vice versa, what is the evidence for the actual absence of nucleotide under no-nucleotide conditions (Figure 2; densities marked with arrowheads)? Clarification of these points is essential for the conclusion '[that] Mg-ADP release ... primes the switch loops for Mg-ATP'

How much light can the data really shine on 'force generation'? Beyond acknowledging that neck linker docking does not provide enough energy for force generation, could the author speculate on where the force is coming from? How do the acquired data relate to the concept of 'electrostatically guided, biased diffusion' (see: Grant, Cross et. al, PLOS Biology 2011, Electrostatically Biased Binding of Kinesin to Microtubules)? Conformational changes could prime the biased binding, but the nucleotide state would merely gate the interactions (as opposed to provide the energy).

---

## [Author Response]

Movement of h6 and neck linker:

*In the Results section it is argued that 'the neck linker remains directed towards the MT minus end...' under no-nucleotide conditions (Figure 3). In the figure, however, it is hard see electron density supporting this statement. In fact, in none of the Figure 3, Figure 3—figure supplement 1) can electron densities be unambiguously detected, corresponding to the neck linker. Figure 3—figure supplement 1 might be an exception but the density extending from h6 could also be attributed to the N-terminal extension (yellow), as seemingly the case for KIF1A (Figure 3—figure supplement 1). The data basis for interpretations on the position of the neck linker is too weak to allow such far-reaching conclusions. The neck linker orientation might be determined using label such as [69]. ([69], The EMBO Journal Vol. 22 No. 7 pp. 1518-1528, 2003)*.

We apologise for the imprecision of the text in this section, which we have adjusted in response to this comment. The major point (which we have re-emphasised in the text) is that in the ADP/NN reconstructions it is clear that the neck linker is *undocked***,** as a result of the relative conformations of helicesα4/α6. Although we can also see that in these reconstructions, the neck linker *emerges* from the motor domain in the direction of the MT minus end, density corresponding to neck linker is only visible for a few amino acids at its N-terminus; the rest of the neck linker is invisible, presumably due to flexibility. The comparison between undocked and docked neck linker – which also emphasises the accompanying change in helix-α6 – is now shown more clearly in the alternative view in the new Figure 3—figure supplement 3. We have truncated the neck linker in the relevant models in Figure 3 and the new figure to more accurately represent the EM density. To enable comparison of density corresponding to the N-terminus vs the neck linker in the AMPPNP/ADPAlFX states, Figure 4 and its supplement provide useful alternative views and we now also refer to these figures. Our data are consistent with the findings of [69] (to which we now refer). Even with their elegant approach of inserting a genetically encoded label (an engineered SH3 domain) onto the C-terminus of the Kin1 neck linker, in monomers this domain was invisible due to neck linker flexibility when undocked.

*The conclusion that 'the beta-sheet tilting ... is not sufficient to describe the conformational changes that accompany force generation...' is an important key conclusion, substantiated by the thorough analysis of alterations in the core motor domain (i.e. without neck linker)*.

*You write (in the Results section) '... no density corresponding to [K-loop and CTT]..., nor ... any effect of motor domain nucleotide state...', and conclude: '... hard to reconcile with ... a specific role of L12 ... in the Mg-ADP state.' This logic is incorrect. The method used (cryo EM reconstruction) does not allow detection of any differences but it cannot be concluded that there ARE no differences (that might be detected by other methods). There may be differences in L12/CTT in dependence on the nucleotide state, just invisible to cryo EM*.

We apologise for this incorrect wording. We have entirely rewritten this section of the text (in the Results section) to address this and the other comments relating to this aspect of our work, discussed further below.

*Furthermore, in the Results section you state '[the ADP-dependent Kin3 motor and track] interaction is not important (or does not exist)...' As argued above, the kinetic data in*
Table 3
*clearly hints at a tight, possibly ionic interaction. Likewise, a '10-fold affinity enhancement cannot [account for] super-processivity' is incorrect. If both heads of a dimeric motor were uncoupled (which they are not, but for the sake of the argument this is irrelevant) and each had a microtubule off-rate of 1/100 s-1, the probability for the dimer to have both heads in the weakly bound state would be (1/100)^2 = 1/10000, which would explain 'super-processivity'. In general, 'super-processive' is a concept that cannot be tested or substantiated in a study with monomeric kinesin motor domain constructs. Speculations on why KIF1A dimers show long run lengths based on structural studies on monomers are highly hypothetical. At least, one would have to know whether 8-nm steps occur, and whether they are coupled 1:1 to ATP hydrolysis*.

The rewritten text in the Results section more clearly and accurately places our structural work in the context of the literature concerning Kin3 biochemistry and mechanism. In brief, a series of landmark studies from the Hirokawa group (38; 52) established the importance of the K-loop in enhancing the interaction of Kin3 monomers with MTs using steady-state ATPase assays and measurement of MT affinity of specific nucleotide-bound states; they also showed that this interaction was ionic strength sensitive. However, they also demonstrated – using mutagenesis and protein engineering - that manipulations of the K-loop alone were not sufficient to account for the >100-fold differences in MT association between Kin3 and Kin1: in these experiments, Kin1/Kin3 constructs ± K-loop showed only 10-fold differences in MT association. Subsequent studies including very recent work from the Verhey group (71; 72) have expanded on these conclusions. In particular, it has been shown that manipulation/deletion of the K-loop does not alter the “super-processivity” of Kin3 dimers in vitro or *in vivo*. Our data cannot provide direct information about the stepping mechanisms of kinesin dimers – and in particular, many factors can contribute to the extent of processivity. However, our work does highlight the structural differences that exist between transport kinesins due to intrinsic properties of the monomeric motor domains that are likely to contribute to differences in motor functionality.

*In the Results section you also state '...the apparent affinity of Kin3 is ∼250x higher than Kin1', and refers to*Table 3*. This table, however, reports catalytic parameters, not affinities*.

*This issue is important for the interpretation of the role of the K-loop in KIF1A. Co-sedimentation assays at varying ionic strengths can reveal the relative importance of ionic and other interactions, possibly in combination with mutants*.

We apologise for the imprecision of this text and have rewritten the relevant section to address these and the related comments . We now use the correct KmMT nomenclature to describe our data. We have also responded by more accurately referencing the rich literature on monomer affinity (detailed above and which already includes binding assays), and the relationship between KmMT and monomer affinity (particularly in [86]). In this context, we agree that dissection of residues that contribute to affinity differences using mutagenesis and cosedimentation assays is an interesting future direction but feel that this is beyond the scope of the current study.

*That KIF1A's observed half-maximal activation constant for microtubules (*Table 3*) is heavily affected by ionic or unspecific interactions can be seen by calculating the kcat/K0.5MT ∼800 s-1 uM-1, which is unrealistically high for a diffusion-limited reaction. The authors have to clearly distinguish between affinity and half-maximal activation constant*.

As described above, we have rewritten the text to correct this lack of clarity with respect to affinity and half-maximal activation constant. As we have also now clarified in the text, it is well described in the literature (in publications from the Hirokawa group in particular) that the interaction of Kif1A monomers with microtubules is heavily affected by ionic/non-specific interactions that lead to the high apparent kcat/K0.5MT values observed.

*What is the evidence for the actual presence of Mg-ADP in the nucleotide binding pocket under ADP conditions?*
Figure 2
*shows KIF1A under this condition but due to the projection into 2D it is not visible whether there is a clear density signal corresponding to the indicated position of Mg-ADP. In this respect,*
Figure 2
*is much less clear than*
Figure 2
*(ADP/AlFx)*.

*Vice versa, what is the evidence for the actual absence of nucleotide under no-nucleotide conditions (*Figure 2*; densities marked with arrowheads)? Clarification of these points is essential for the conclusion '[that] Mg-ADP release ... primes the switch loops for Mg-ATP'*

We agree that this is a critical point: 1) We have added additional, equivalent views of our reconstructions that demonstrate nucleotide occupancy in the new Figure 2—figure supplement 5, which is now referred to in the Results text. 2) We have provided additional references in the Materials and methods section, reinforcing that our sample preparation was based on established protocols. 3) Overall motor domain conformations (including neck-linker orientation) are consistent with nucleotide assignment as expected from the literature.

*How much light can the data really shine on 'force generation'? Beyond acknowledging that neck linker docking does not provide enough energy for force generation, could the author speculate on where the force is coming from? How do the acquired data relate to the concept of 'electrostatically guided, biased diffusion' (see: Grant, Cross et. al, PLoS Biology 2011, Electrostatically Biased Binding of Kinesin to Microtubules)? Conformational changes could prime the biased binding, but the nucleotide state would merely gate the interactions (as opposed to provide the energy)*.

The major contribution of our data to understanding force generation is to show that the nucleotide-dependent conformational changes in MT-bound Kin1 and Kin3 are conserved. This is particularly significant given that alternative and quite distinct models for Kin3 force generation have previously been proposed. The nucleotide-dependent conformations captured in our reconstructions presumably occur after any biased diffusion – as described in the interesting study of Grant et al – has ceased. By their nature, the structure(s) of the motors as they undergo biased diffusion would not be accessible using our averaging methods. In contrast, the conformational changes that we describe depend on the stereospecific interaction of the motors with the MT and are preferentially stabilised by the particular nucleotide bound. The point(s) in the MT-bound ATPase cycle where force is generated and how the neck linker contributes to this remain somewhat disputed in the field, and we have altered the text throughout the manuscript to more accurately reflect this. Our higher resolution structures provide snapshots of the motor domain and thus a mechanistic framework in which hypotheses concerning force generation can be considered. We have added some more speculative text on this point in the Results section.